# PAPR: Proximity Attention Point Rendering

**Yanshu Zhang**\*, **Shichong Peng**\*, **Alireza Moazeni**, **Ke Li**
APEX Lab
School of Computing Science
Simon Fraser University
{yanshu_zhang,shichong_peng,seyed_alireza_moazenipourasil,keli}@sfu.ca

## Abstract

Learning accurate and parsimonious point cloud representations of scene surfaces from scratch remains a challenge in 3D representation learning. Existing point-based methods often suffer from the vanishing gradient problem or require a large number of points to accurately model scene geometry and texture. To address these limitations, we propose Proximity Attention Point Rendering (PAPR), a novel method that consists of a point-based scene representation and a differentiable renderer. Our scene representation uses a point cloud where each point is characterized by its spatial position, influence score, and view-independent feature vector. The renderer selects the relevant points for each ray and produces accurate colours using their associated features. PAPR effectively learns point cloud positions to represent the correct scene geometry, even when the initialization drastically differs from the target geometry. Notably, our method captures fine texture details while using only a parsimonious set of points. We also demonstrate four practical applications of our method: zero-shot geometry editing, object manipulation, texture transfer, and exposure control. More results and code are available on our project website.

## 1   Introduction

Learning 3D representations is crucial for computer vision and graphics applications. Recent neural rendering methods [22, 42, 50, 23, 15, 48] leverage deep neural networks within the rendering pipelines to capture complex geometry and texture details from multi-view RGB images. These methods have many practical applications, such as generating high-quality 3D models [18, 43, 3, 7] and enabling interactive virtual reality experiences [29, 25, 52]. However, balancing representation capacity and computational complexity remains a challenge. Large-scale representations offer better quality but demand more resources. In contrast, parsimonious representations strike a balance between efficiency and quality, enabling efficient learning, processing, and manipulation of 3D data.

3D representations can be categorized into volumetric and surface representations. Recent advances in volumetric representations [22, 48, 50, 37, 17] have shown impressive results in rendering quality. However, the cubic growth in encoded information as the scene radius increases makes volumetric representations computationally expensive to process. For example, to render a volumetric representation, all the information along a ray needs to be aggregated, which requires evaluating many samples along each ray. In contrast, surface representations are more parsimonious and efficient since the encoded information grows quadratically. So surface representations can be rendered with much fewer samples.

Efficient surface representations include meshes and surface point clouds. Meshes are difficult to learn from scratch, since many constraints (e.g., no self-intersection) need to be enforced. Additionally, the topology of the initial mesh is fixed, making it impossible to change during training [43]. In contrast,

---

\*Denotes equal contribution

37th Conference on Neural Information Processing Systems (NeurIPS 2023).

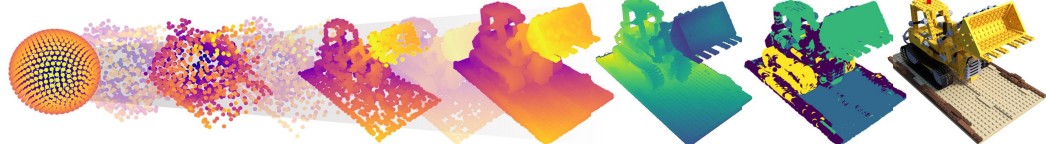

(a) Point Cloud Position      (b) Depth    (c) Point Feature (d) Rendering

Figure 1: Our proposed method, PAPR, jointly learns a point-based scene representation and a differentiable renderer from scratch using only RGB supervision. PAPR effectively learns the point positions (a) that represent the correct surface geometry, as demonstrated by the depth map (b). Notably, PAPR achieves this even when starting from an initialization that substantially deviates from the target geometry. Furthermore, PAPR learns a view-independent feature vector for each point, capturing the local scene content. The clustering of point feature vectors is shown in (c). The renderer selects and combines these feature vectors to generate high-quality colour rendering (d).

point clouds offer more flexibility in modelling shapes with varying topologies. Hence, in this work, we focus on using point clouds to represent scene surfaces.

Learning a scene representation from scratch requires a differentiable renderer that can pass large gradients to the representation. Designing a differentiable renderer for point clouds is non-trivial – each point is infinitesimal and rarely intersects with rays, it is unclear what the output colour at such rays should be. Previous methods often rely on splat-based rasterization techniques, where points are turned into disks or spheres [45, 15, 35, 53, 11, 56]. These methods use a radial basis function (RBF) kernel to calculate the contribution of each point to each pixel. However, determining the optimal radius for the splats or the RBF kernel is challenging [9], and these methods suffer from the issue of vanishing gradient when the ray is far away from the points, limiting their ability to learn the ground truth geometry that drastically differs from the initial geometry. Additionally, the radius of each splat should be small for accurate texture modelling, and this would require a point cloud with a large number of points, which are difficult to process.

To address these limitations, we introduce a novel method called Proximity Attention Point Rendering (PAPR). PAPR consists of a point-based scene representation and a differentiable renderer. The scene representation is constructed using a point cloud, where each point is defined by its spatial position, an influence score indicating its influence on the rendering, and a view-independent feature vector that captures the local scene content. Our renderer learns to directly select points for each ray and combines them to generate the correct colour using their associated features. Remarkably, PAPR can learn accurate geometry and texture details using a parsimonious set of points, even when the initial point cloud substantially differs from the target geometry, as shown in Figure 1. We show that the proposed ray-dependent point embedding design is crucial for effective learning of point cloud positions from scratch. Furthermore, our experiments on both synthetic and real-world datasets demonstrate that PAPR outperforms prior point-based methods in terms of image quality when using a parsimonious set of points. We also showcase four practical applications of PAPR: zero-shot geometry editing with part rotations and deformations, object duplication and deletion, texture transfer and exposure control. In summary, our contributions are as follows:

1. We propose PAPR, a novel point-based scene representation and rendering method that can learn point-based surface geometry from scratch.

2. We demonstrate PAPR's ability to capture correct scene geometry and accurate texture details using a parsimonious set of points, leading to an efficient and effective representation.

3. We explore and demonstrate four practical applications with PAPR: zero-shot geometry editing, object manipulation, texture transfer and exposure control.

## 2 Related Work

Our work focuses on learning and rendering 3D representation, and it is most related to differentiable rendering, neural rendering and point-based rendering. While there are many relevant works in these fields, we only discuss the most pertinent ones in this context and refer readers to recent surveys [9, 41, 47] for a more comprehensive overview.

**Differentiable Rendering For 3D Representation Learning**    Differentiable rendering plays a crucial role in learning 3D representations from data as it allows gradients to be backpropagated from the rendered output to the representation. Early mesh learning approach, OpenDR [19], approximate gradients using localized Taylor expansion and differentiable filters, but do not fully leverage loss function gradients. Neural Mesh Renderer [10] proposes non-local approximated gradients that utilize gradients backpropagated from a loss function. Some methods [33, 18, 27] modify the forward rasterization step to make the pipeline differentiable by softening object boundaries or making the contribution of triangles to each pixel probabilistic. Several point cloud learning methods, such as differentiable surface splatting [44] and differentiable Poisson solvers [26], explore surface reconstruction techniques but they do not focus on rendering fine textures. In contrast, our method jointly learns detailed colour appearance and geometry representations from scratch.

**Neural Scene Representation**    In recent years, there has been a growing interest in utilizing neural networks to generate realistic novel views of scenes. Some approaches utilize explicit representations like meshes [42, 49], multi-plane images [5, 20, 21, 36, 55], and point clouds [1, 30, 15]. Another approach represents the scene as a differentiable density field, as demonstrated by Neural Radiance Field (NeRF) [22]. However, NeRF suffers from slow sampling speed due to volumetric ray marching. Subsequent works have focused on improving training and rendering speed through space discretization [39, 6, 8, 4, 31, 51, 46] and storage optimizations [23, 38, 50, 37, 17]. These methods still rely on volumetric representations, which have a cubic growth in encoded information. In contrast, our proposed method leverages surface representations, which are more efficient and parsimonious.

**Point-based Representation Rendering and Learning**    Rendering point clouds using ray tracing is challenging because they do not inherently define a surface, leading to an ill-defined intersection between rays and the point cloud. Early approaches used splatting techniques with disks, ellipsoids, or surfels [57, 32, 2, 28]. Recent methods [1, 30, 24, 14] incorporate neural networks in the rendering pipeline but assume ground truth point clouds from Structure-from-Motion (SfM), Multi-View Stereo (MVS) or LiDAR scanning without optimizing point positions. To learn the point positions, splat-based rasterization methods [45, 15, 35, 53, 11, 56] employ radial basis function kernels to compute point contributions to pixels but struggle with finding optimal splat or kernel radius [9] and suffer from gradient vanishing when points are far from target pixels. Consequently, these methods require a large number of points to accurately model scene details and are limited to learn small point cloud deformation. In contrast, our method can capture scene details using a parsimonious set of points and perform substantial deformation in the initial point cloud to match the correct geometry.

Another approach [3] predicts explicit intersection points between rays and optimal meshes reconstructed from point clouds, but it requires auxiliary data with ground truth geometry for learning and RGBD images for point cloud initialization. Our method overcomes these limitations. Xu et al. [48] represent radiance fields with points and use volumetric ray marching, our approach utilizes more efficient and parsimonious surface representations and generates output colour for each ray with only a single forward pass through the network.

## 3   Proximity Attention Point Rendering

Figure 2 provides an overview of our proposed end-to-end learnable point-based rendering method. The input to our model includes a set of RGB images along with their corresponding camera intrinsics and extrinsics. Our method jointly learns a point-based scene representation, where each point comprises a spatial position, an influence score, and a view-independent feature vector, along with a differentiable renderer. This is achieved by minimizing the reconstruction loss between the rendered image $\hat{\mathbf{I}}$ and the ground truth image $\mathbf{I}_{gt}$ according to some distance metric $d(\cdot, \cdot)$, w.r.t. the scene representation and the parameters of the differentiable renderer. Unlike previous point cloud learning methods [48, 45, 15, 35, 53, 11, 3, 56] which rely on multi-view stereo (MVS), Structure-from-Motion (SfM), depth measurements or object masks for initializing the point cloud positions, our method can learn the point positions from scratch, even when the initialization significantly differs from the target geometry as demonstrated in Figure 4. In the following sections, we provide a more detailed description of each component of our proposed method.

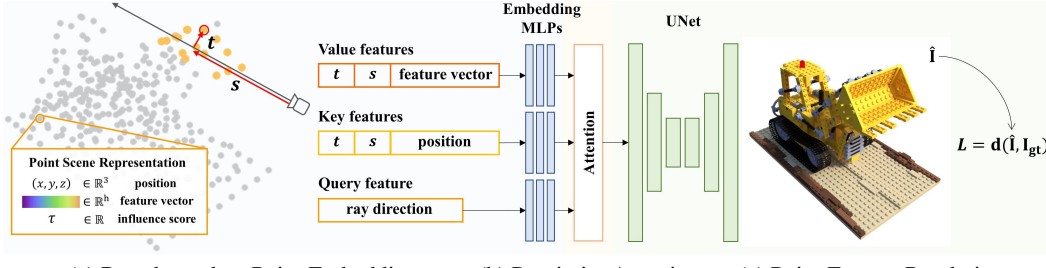

(a) Ray-dependent Point Embedding     (b) Proximity Attention     (c) Point Feature Rendering

Figure 2: An overview of the proposed pipeline for rendering a point-based scene representation, where each point is defined by its spatial position, an influence score, and a view-independent feature vector. (a) Given a ray, ray-dependent features are created for each point. These features are used to generate the key, value, and query inputs for a proximity attention model. (b) The attention model selects the points based on their keys and the query, and combines their values to form an aggregated feature. (c) The aggregated feature is gathered for all pixels to create a feature map. This feature map is then passed through a feature renderer, implemented using a UNet architecture, to produce the output image. The entire pipeline is jointly trained in an end-to-end manner using only supervision from the ground truth image.

## 3.1 Point-based Scene Representation

Our scene representation consists of a set of $N$ points $P = \{(\mathbf{p}_i, \mathbf{u}_i, \tau_i)\}_{i=1}^{N}$ where each point $i$ is characterized by its spatial position $\mathbf{p}_i \in \mathbb{R}^3$, an individual view-independent feature vector $\mathbf{u}_i \in \mathbb{R}^h$ that encodes local appearance and geometry and an influence score $\tau_i \in \mathbb{R}$ that represents the influence of the point on the rendering.

Notably, we make the feature vector for each point to be independently learnable, thereby allowing the content encoded in the features to be independent of the distance between neighbouring points. As a result, after training, points can be moved to different positions without needing to make a corresponding change in the features. This design also makes it possible to increase the fidelity of the representation by adding more points, thereby capturing more fine-grained details. Additionally, we specifically choose the feature vectors to be view-independent, ensuring that the encoded information in the representation remains invariant to view changes.

## 3.2 Differentiable Point Rendering with Relative Distances

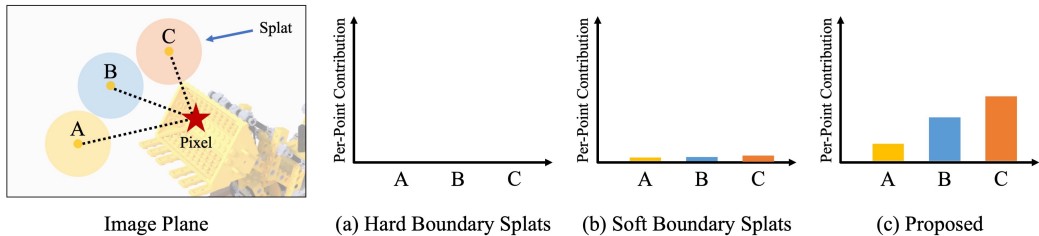

Image Plane          (a) Hard Boundary Splats     (b) Soft Boundary Splats     (c) Proposed

Figure 3: Comparison of point contributions in splat-based rasterization and the proposed method. Three points are projected onto the image plane, with the ground truth image shown in the background. The bar charts illustrate the contribution of each point towards a specific pixel, represented by the red star. (a) If the points are splats with hard boundaries, none of them intersect with the given pixel, resulting in zero contributions. Consequently, the gradient of the loss at the pixel w.r.t. each point is non-existent, hindering the reduction of loss at that pixel. (b) For splats with soft boundaries, although the point contributions are non-zero, they are extremely small, leading to vanishing gradients. (c) Our proposed method normalizes the contribution of all points to the given pixel. This ensures that there are always points with substantial contributions, enabling learning point positions from scratch.

To learn the scene representation from images from different views, we minimize the reconstruction loss between the rendered output from those views and the ground truth image w.r.t. the scene representation. Typically, we do so with gradient-based optimization methods, which need to compute

the gradient of the loss w.r.t. the scene representation. For this to be possible, the rendered output must be differentiable w.r.t. the representation – in other words, the renderer must be differentiable.

In classical rendering pipelines, point clouds are often rendered with splat-based rasterization. However, when attempting to learn point clouds from scratch using this method, several challenges arise for the following reasons. A splat only contributes to the pixels it intersects with, so when a splat does not intersect with a specific pixel, it does not contribute to the rendered output at that pixel. As a result, the gradient of the loss at that pixel w.r.t. the position of the splat is zero. Now, consider a pixel in the ground truth image that no splat intersects with. Even though the loss at that pixel might be high, the gradient w.r.t. the position of every splat is zero, so the loss at that pixel cannot be minimized, as illustrated in Figure 3a.

To address this issue, prior methods have employed a radial basis function (RBF) kernel to soften splat boundaries. This approach makes the contribution of each splat to each pixel non-zero. However, even with the soft boundaries, the contribution of a splat to a pixel remains small if the pixel is located far away from the centre of the splat. Consequently, for a pixel far away from the centres of all splats, the gradient of the loss at the pixel w.r.t. each splat remains too minuscule to exert a significant influence on its position, as shown in Figure 3b – in other words, the gradient vanishes. As a result, even though the loss at the pixel might be high, it will take a long time before it is minimized.

We observe that with both hard- and soft-boundary splats, the contribution of each point to a given pixel only depends on its own *absolute distance* to the pixel, without regard to the distances of other points. This gives rise to the vanishing gradient problem because the absolute distances for all points can be high. To address this issue, we make the contributions per point towards a pixel depend on *relative* distances rather than *absolute* distances. Instead of relying solely on the distance to the pixel from a single point to compute its contribution, we normalize the contributions across all points so that the total contribution from all points is always 1. This ensures that there are always points with significant contributions to each pixel, even if the pixel is far away from all points, as shown in Figure 3c.

This allows points to be moved far from their initial positions, so the initialization can differ substantially from the ground truth scene geometry. This also means that we can move sufficiently many points from elsewhere to represent the scene geometry faithfully. On the other hand, the learnt representation uses no more points than necessary to represent each local part of the scene geometry faithfully, because if there were extraneous points, they could be moved to better represent other parts. The end result is a parsimonious scene representation that uses just enough points to faithfully represent each part of the scene geometry. So, fewer points tend to be allocated to smooth parts of the surface and the interior of opaque surfaces, because having more points there would not improve the rendering quality significantly. Moreover, since the points can be far away from a pixel without having their contributions diminished, we can preserve the surface continuity even if we apply a non-volume preserving transformation (e.g., stretching) to the point cloud post-hoc.

### 3.3 Implementation of Relative Distances with Proximity Attention

We design an attention mechanism to compute contributions from relative distances, which we dub *proximity attention*. Unlike typical attention mechanisms, the queries correspond to rays and the keys correspond to points. To encode the relationship between points and viewing directions, we design an embedding of the points that account for their relative positions to rays. This embedding is then fed in as input to the proximity attention mechanism. We describe the details of each below.

**Ray-dependent Point Embedding**   Given a camera pose $\mathcal{C}$ specified by its extrinsics and intrinsics, and a sensor resolution $H \times W$, we utilize a pinhole camera model to cast a ray $\mathbf{r}_j$ from the camera centre to each pixel. Each ray $\mathbf{r}_j$ is characterized by its origin $\mathbf{o}_j \in \mathbb{R}^3$ and unit-length viewing direction $\mathbf{d}_j \in \mathbb{R}^3$, both defined in the world coordinate system.

To obtain the ray-dependent point embedding of point $\mathbf{p}_i$ with respect to ray $\mathbf{r}_j$, we start by finding the projection $\mathbf{p}'_i$ of the point onto the ray:

$$\mathbf{p}'_i = \mathbf{o}_j + \langle \mathbf{p}_i - \mathbf{o}_j, \mathbf{d}_j \rangle \cdot \mathbf{d}_j \tag{1}$$

where '$\langle \cdot, \cdot \rangle$' denotes the inner product. Next, we compute the displacement vectors, $\mathbf{s}_{i,j}$ and $\mathbf{t}_{i,j}$:

$$\mathbf{s}_{i,j} = \mathbf{p}'_i - \mathbf{o}_j, \ \mathbf{t}_{i,j} = \mathbf{p}_i - \mathbf{p}'_i, \tag{2}$$

Here, $\mathbf{s}_{i,j}$ captures the depth of the point $\mathbf{p}_i$ with respect to the camera centre $\mathbf{o}_j$ along the ray, while $\mathbf{t}_{i,j}$ represents the perpendicular displacement from the ray to the point $\mathbf{p}_i$. The final ray-dependent point embedding contains these displacement vectors, $\mathbf{s}_{i,j}$ and $\mathbf{t}_{i,j}$, as well as the point position $\mathbf{p}_i$.

**Proximity Attention**    For a given ray $\mathbf{r}_j$, we start by filtering the top $K$ nearest points $\mathbf{p}_i$ based on the magnitude of their perpendicular displacement vector $||\mathbf{t}_{i,j}||$. This allows us to identify the points that are most relevant for modelling the local contents of the ray.

We adopt the ray-dependent point embedding method described earlier for the $K$ closest points, which serves as the input to construct the key in the attention mechanism. For the value branch input, we utilize the $K$ associated point feature vectors $\mathbf{u}_i$, along with their displacement vectors $\mathbf{s}_{i,j}$ and $\mathbf{t}_{i,j}$. For the input to the query branch, we use the ray direction $\mathbf{d}_j$. Additionally, we apply positional encoding [22, 40] $\gamma(p) = \left[\sin\left(2^l \pi p\right), \cos\left(2^l \pi p\right)\right]_{l=0}^{L=6}$ to all components except for the feature vector. These inputs pass through three separate embedding networks $f_{\theta_K}$, $f_{\theta_V}$ and $f_{\theta_Q}$, which are implemented as multi-layer perceptrons (MLPs). The resulting outputs form the final key, value, and query for the attention mechanism:

$$\mathbf{k}_{i,j} = f_{\theta_K}\left([\gamma\left(\mathbf{s}_{i,j}\right), \gamma\left(\mathbf{t}_{i,j}\right), \gamma\left(\mathbf{p}_i\right)]\right) \tag{3}$$

$$\mathbf{v}_{i,j} = f_{\theta_V}\left([\gamma\left(\mathbf{s}_{i,j}\right), \gamma\left(\mathbf{t}_{i,j}\right), \mathbf{u}_i]\right) \tag{4}$$

$$\mathbf{q}_j = f_{\theta_Q}\left(\gamma\left(\mathbf{d}_j\right)\right) \tag{5}$$

To compute the aggregated value for the query, we calculate the weighted sum of the values as $\mathbf{f}_j = \sum_{i=1}^K w_{i,j}\mathbf{v}_{i,j}$, where each weighting term is calculated by applying the softmax function to the raw attention scores $a_{i,j}$. The raw attention score $a_{i,j}$ is obtained by applying a ReLU function to the dot product between the query $\mathbf{q}_j$ and the keys $\mathbf{k}_{i,j}$ scaled by their dimensionality $\mu$:

$$w_{i,j} = \frac{\exp(a_{i,j})}{\sum_{m=1}^K \exp(a_{m,j})}, \text{ where } a_{i,j} = \max(0, \frac{\langle \mathbf{q}_j, \mathbf{k}_{i,j} \rangle}{\sqrt{\mu}}) \tag{6}$$

By aggregating the features $\mathbf{f}_j$ for all rays $\mathbf{r}_j$ corresponding to each pixel on the image plane with resolution $H \times W$, we construct the feature map $F_{\mathcal{C}} \in \mathbb{R}^{H \times W \times d_{\text{feat}}}$ for a given camera pose $\mathcal{C}$.

**Point Feature Renderer**    To obtain the final rendered colour output image $\hat{\mathbf{I}}$ for a given camera pose $\mathcal{C}$, we input the aggregated feature map to a convolutional feature renderer, $\hat{\mathbf{I}} = f_{\theta_{\mathcal{R}}}(F_{\mathcal{C}})$. Our renderer is based on a modified version of the U-Net architecture [34], featuring two downsampling layers and two upsampling layers. Notably, we remove the BatchNorm layers in this network. For more details regarding the model architecture, please refer to the appendix.

### 3.4   Progressive Refinement of Scene Representation

**Point Pruning**    To eliminate potential outliers during the point position optimization process, we introduce a point pruning strategy that utilizes a learnable influence score $\tau_i \in \mathbb{R}$ associated with each point. We calculate the probability $w_{b,j} \in [0, 1]$ of a ray $\mathbf{r}_j$ hitting the background by comparing a fixed background token $b$ to the product of each point's influence score $\tau_i$ and its raw attention score $a_{i,j}$ (defined in Eqn. 6). The probability is given by:

$$w_{b,j} = \frac{\exp(b)}{\exp(b) + \sum_{m=1}^K \exp(a_{m,j} \cdot \tau_m)} \tag{7}$$

By collecting the background probabilities $w_{b,j}$ for all $H \times W$ rays cast from camera pose $\mathcal{C}$, we obtain a background probability mask $P_{\mathcal{C}} \in \mathbb{R}^{H \times W \times 1}$. The rendered output image $\hat{\mathbf{I}}$ can then be computed using this background probability mask as follows:

$$\hat{\mathbf{I}} = (1. - P_{\mathcal{C}}) \cdot f_{\theta_{\mathcal{R}}}(F_{\mathcal{C}}) + P_{\mathcal{C}} \cdot \mathbf{I}_b \tag{8}$$

Here, $\mathbf{I}_b \in \mathbb{R}^{H \times W \times 3}$ represents the given background colour. All influence scores are initialized to zero. Starting from iteration $10,000$, we prune points with $\tau_i < 0$ every 500 iterations. Additional details and comparisons to the model without pruning are available in the appendix.

**Point Growing**    As mentioned in Sec. 3.1, our scene representation allows for increased modelling capacity by adding more points. To achieve this, we propose a point growing strategy that incrementally adds points to the sparser regions of the point cloud every $500$ iterations until the desired total number of points is reached. More information on how the sparse regions are identified can be found in the appendix. Fig. 1a illustrates the effectiveness of our point growing strategy in progressively refining the point cloud, resulting in a more detailed representation of the scene.

## 3.5    Training Details

Our training objective is minimizing the distance metric $d(\cdot, \cdot)$ from the rendered image $\hat{\mathbf{I}}$ to the corresponding ground truth image $\mathbf{I}_{gt}$. We define the distance metric as a weighted combination of mean squared error (MSE) and the LPIPS metric [54]. The loss function is formulated as:

$$\mathcal{L} = d(\hat{\mathbf{I}}, \mathbf{I}_{gt}) = \text{MSE}(\hat{\mathbf{I}}, \mathbf{I}_{gt}) + \lambda \cdot \text{LPIPS}(\hat{\mathbf{I}}, \mathbf{I}_{gt}) \tag{9}$$

We set the weight $\lambda$ to $0.01$ for all experiments. During training, we jointly optimize all model parameters, including $\mathbf{p}_i, \mathbf{u}_i, \tau_i, \theta_K, \theta_V, \theta_Q$ and $\theta_{\mathcal{R}}$. We train our model using Adam optimizer [12] on a single NVIDIA A100 GPU.

## 3.6    Learning Point Positions From Scratch

We demonstrate the effectiveness of our method in learning point positions from scratch using the Lego scene in the NeRF Synthetic Dataset [22]. We compare our proposed method to recent point-based methods, namely DPBRF [53], Point-NeRF [48], NPLF [24] and Gaussian Splatting [11]. It is worth noting that NPLF does not inherently support point position learning, so we made modifications to their official code to enable point position optimization.

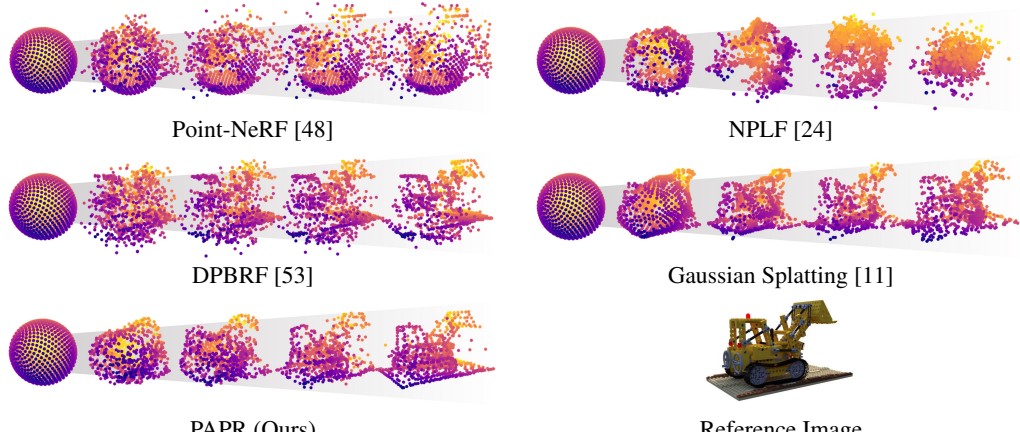

Figure 4: Comparison of our method and prior point-based methods on learning point cloud positions from scratch. We modify NPLF [24]'s official implementation to support point position updates. All methods start with the same initial point cloud of $1,000$ points on a sphere. Point pruning and growing strategies are disabled for a fair comparison of geometry learnability. An RGB image of the scene from the same view point as the point cloud visualizations is provided as a reference. Point-NeRF [48] and NPLF [24] fail to reconstruct a reasonable point cloud. DPBRF [53] captures a rough shape but introduces significant noise and struggles to capture the rear end of the Lego bulldozer effectively. Gaussian Splatting [11] captures the Lego silhouette but lacks structural details, such as the bulldozer's track (the black belt around the wheels). In contrast, our method successfully learns a point cloud that accurately represents the object's surface and captures detailed scene structures.

In our comparison, all methods start with the same set of $1,000$ points initialized on a sphere. To ensure a fair evaluation of point cloud position learnability, we disable point pruning and growing techniques for all methods. As shown in Figure 4, our method successfully deforms the initial point cloud to correctly represent the target geometry. In contrast, the baselines either fail to recover the geometry, produce noisy results, or lack structural details in the learnt geometry.

# 4 Experiments

To validate our contribution of learning point-based scene representation and rendering pipeline directly from multi-view images, we compare our method to recent point-based neural rendering methods, namely NPLF [24], DPBRF [53], SNP [56], Point-NeRF [48] and Gaussian Splatting [11]. As a secondary comparison, we demonstrate the potential broader impact of our method by comparing it to the widely used implicit volumetric representation method NeRF [22].

To assess the performance of scene modelling with a parsimonious representation, we conduct evaluations using a total of $30,000$ points for each method. For all point-based baselines, we use their original point cloud initialization methods, which are based on Multi-View Stereo (MVS), Structure-from-Motion (SfM) or visual hull. Conversely, for our approach, we adopt a random point cloud initialization strategy within a predefined volume, introducing an additional level of complexity and challenge to our method. We set the parameter $K = 20$ for selecting the top nearest points, and the point feature vector dimension $h = 64$. We evaluate all methods in both synthetic and real-world scenarios. For the synthetic setting, we choose the NeRF Synthetic dataset [22], while for the real-world setting, we use the Tanks & Temples [13] subset, following the same data pre-processing steps as in [48]. We evaluate rendered image quality using PSNR, SSIM and LPIPS [54] metrics.

## 4.1 Quantitative Results

Table 1 shows the average image quality metric scores. PAPR consistently outperforms the baselines across all metrics in both synthetic and real-world settings, without relying on specific initialization. These results demonstrate PAPR's ability to render highly realistic images that accurately capture details using an efficient and parsimonious representation.

| | NeRF Synthetic | | | Tank & Temples | | | |
|---|---|---|---|---|---|---|---|
| | PSNR ↑ | SSIM ↑ | LPIPS$_{vgg}$ ↓ | PSNR ↑ | SSIM ↑ | LPIPS$_{vgg}$ ↓ | Initialization |
| NeRF [22] | 31.00 | 0.947 | 0.081 | 27.94 | 0.904 | 0.168 | — |
| NPLF [24] | 18.36 | 0.780 | 0.213 | 21.19 | 0.761 | 0.240 | MVS |
| DPBRF [53] | 25.61 | 0.884 | 0.138 | 17.25 | 0.634 | 0.301 | Visual Hull |
| SNP [56] | 26.00 | 0.914 | 0.110 | 26.39 | 0.894 | 0.160 | MVS |
| Point-NeRF [48] | 25.93 | 0.923 | 0.129 | 24.75 | 0.890 | 0.184 | MVS |
| Gaussian Splatting [11] | 27.76 | 0.929 | 0.084 | 26.81 | 0.907 | 0.140 | SfM |
| PAPR (Ours) | **32.07** | **0.971** | **0.038** | **28.72** | **0.940** | **0.097** | Random |

Table 1: Comparison of image quality metrics (PSNR, SSIM and LPIPS [54]) on the NeRF Synthetic dataset [22] and Tanks & Temples subset [13]. Higher PSNR and SSIM scores are better, while lower LPIPS scores are better. All point-based baselines use a total number of $30,000$ points and are initialized using the original techniques proposed by their respective authors. Despite being initialized from scratch, our method outperforms all baselines on all metrics for both datasets with the same total number of points.

## 4.2 Qualitative Results

Figure 5 shows a qualitative comparison between our method, PAPR, and the baselines on the NeRF Synthetic dataset. PAPR produces images with sharper details compared to the baselines. Notably, our method captures fine texture details on the bulldozer's body, the reflections on the drums and the crash, the mesh on the mic, and the reflection on the material ball. In contrast, the baselines either fail to capture the texture, exhibit blurriness in the generated output, or introduce high-frequency noise. These results validate the effectiveness of PAPR in capturing fine details and generating realistic renderings compared to the baselines. For additional qualitative results, please refer to the appendix.

## 4.3 Ablation Study

**Effect of Number of Points** We analyze the impact of the number of points on the rendered image quality in our method. The evaluation is performed using the LPIPS metric on the NeRF synthetic dataset. As shown in Figure 6, our method achieves better image quality by using a higher number of points. Additionally, we compare the performance of our model to that of DPBRF and Point-NeRF. The results show that our model maintains higher rendering quality even with a reduced number of

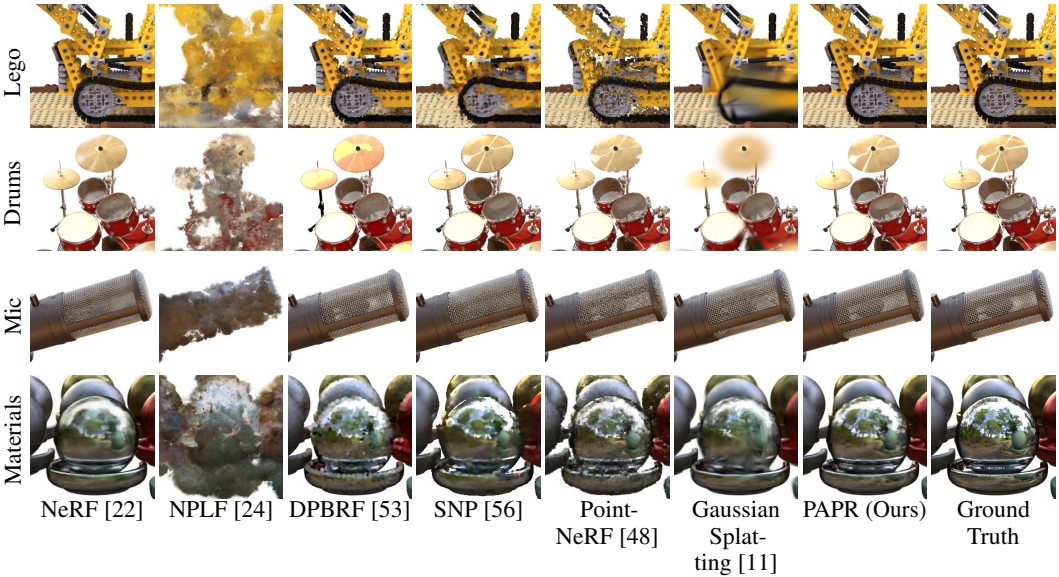

Figure 5: Qualitative comparison of novel view synthesis on the NeRF Synthetic dataset [22]. All point-based methods use a total number of $30,000$ points. NPLF [24] fails to capture the texture detail, NeRF [22], SNP [56] and Gaussian Splatting [11] exhibit blurriness in the rendered images, while DPBRF [53] and Point-NeRF [48] display high-frequency artifacts. In contrast, PAPR achieves high-quality rendering without introducing high-frequency noise, demonstrating its ability to capture fine texture details.

points, as few as 1,000 points, compared to the baselines. These findings highlight the effectiveness of our method in representing the scene with high quality while using a parsimonious set of points.

**Effect of Ray-dependent Point Embedding Design**    We analyze our ray-dependent point embedding design by gradually removing components, including the point position $p_i$ and the displacement vector $t_{i,j}$, which are described in Sec. 3.3. Figure 7 shows both the qualitative and quantitative results of this analysis. Removing $p_i$ in key features introduces increased noise in the learnt point cloud, indicating its importance. Likewise, removing $t_{i,j}$ in both key and value features results in a failure to learn the correct geometry. These findings validate the effectiveness of our point embedding design and highlight the necessity of both the point position and displacement vector for achieving accurate and high-quality results.

## 4.4   Practical Applications

**Zero-shot Geometry Editing**    We showcase the zero-shot geometry editing capability of our method by manipulating the positions of the point representations *only*. In Figure 8a, we present three cases that deform the object geometry: (1) rigid bending of the ficus branch, (2) rotation of the statue's head, and (3) stretching the tip of the microphone to perform a non-volume preserving transformation. The results demonstrate that our model effectively preserves surface continuity after geometry editing, without introducing holes or degrading the rendering quality. Additional results and comparisons with point-based baselines can be found in the appendix.

**Object Manipulation**    In Figure 8b, we present object duplication and removal scenarios. Specifically, we add an additional hot dog to the plate and perform removal operations on some of the balls in the materials scene while duplicating others. These examples demonstrate the versatility of our method in manipulating and editing the objects in the scene.

**Texture Transfer**    We showcase the capability of our method to manipulate the scene texture in Figure 8c. In this example, we demonstrate texture transfer by transferring the associated feature vectors from points corresponding to the mustard to some of the points corresponding to the ketchup. The model successfully transfers the texture of the mustard onto the ketchup, resulting in a realistic and coherent texture transformation. This demonstrates the ability of our method to manipulate and modify the scene texture in a controlled manner.

**Exposure Control**    We showcase the ability of our method to adjust the exposure of the rendered image. To accomplish this, we introduce an additional random latent code input to our feature

renderer $f_{\theta_{\mathcal{R}}}$ and finetune the model using the cIMLE technique [16]. Figure 8d shows different exposures in the rendered image by varying the latent code input. For more detailed information on this application, please refer to the appendix. This demonstrates our method's flexibility in controlling exposure and achieving desired visual effects in the rendered images.

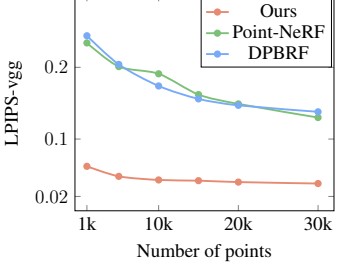

Figure 6: Ablation study on the effect of the number of points on rendered image quality. The results show that increasing the number of points improves the performance of our method. Moreover, our approach consistently outperforms the baselines when using an equal number of points, while also exhibiting greater robustness to a reduction in the number of points.

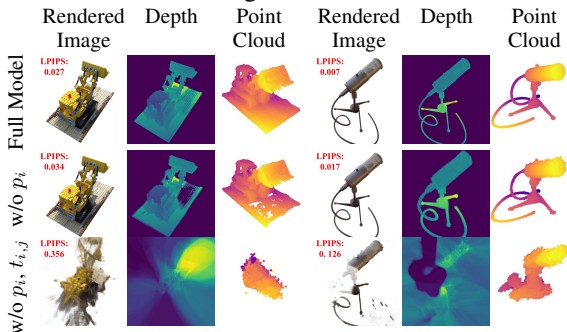

Figure 7: Ablation study on the design for the ray-dependent point embedding. We incrementally remove key components in the point embedding, starting with the point position $p_i$ and then the displacement vector $t_{i,j}$. The results show a significant degradation in the learnt geometry and rendering quality at each step of removal. This validates the importance of each component in the design of our point embedding.

## 5  Discussion and Conclusion

**Limitation**   Our pruning strategy currently assumes the background image to be given, making it suitable for scenarios with a near-constant background colour. However, this assumption may not hold in more diverse and complex backgrounds. To address this limitation, we plan to explore learning a separate model dedicated to handling background variations in future work.

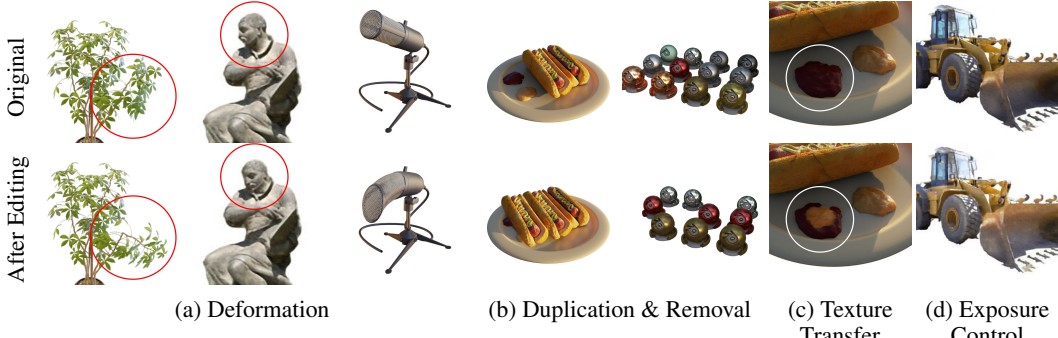

(a) Deformation        (b) Duplication & Removal    (c) Texture Transfer    (d) Exposure Control

Figure 8: Applications of PAPR: (a) Zero-shot Geometry editing - deforming objects (bending branch, rotating head, stretching mic), (b) Object manipulation - duplicating and removing objects, (c) Texture transfer - transferring texture from mustard to ketchup, (d) Exposure control - changing from dark to bright.

**Societal Impact**   Expanding our method's capacity with more points and deeper networks can improve rendering quality. However, it's important to consider the environmental impact of increased computational resources, potentially leading to higher greenhouse gas emissions.

**Conclusion**   In this paper, we present a novel point-based scene representation and rendering method. Our approach overcomes challenges in learning point cloud from scratch and effectively captures correct scene geometry and accurate texture details with a parsimoinous set of points. Furthermore, we demonstrate the practical applications of our method through four compelling use cases.

**Acknowledgements**   This research was enabled in part by support provided by NSERC, the BC DRI Group and the Digital Research Alliance of Canada.

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

# A Implementation Details

## A.1 Proximity Attention

As introduced in Sec. 3.3, our model utilizes an attention mechanism to determine the proximity between the ray $\mathbf{r}_j$ and a point $i$. To achieve this, we use three independent embedding MLPs to compute the key $\mathbf{k}_{i,j}$, value $\mathbf{v}_{i,j}$ and query $\mathbf{q}_j$ for the attention mechanism. The architecture details of these three embedding MLPs are shown in Figure 9.

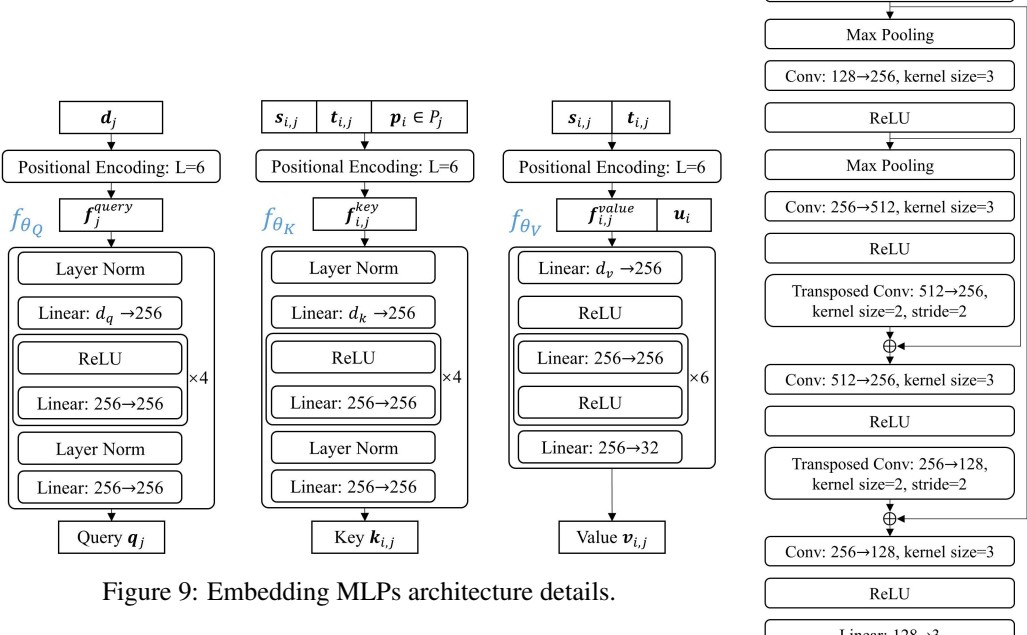

Figure 9: Embedding MLPs architecture details.

Figure 10: U-Net details.

## A.2 Point Feature Renderer

As described in Sec. 3.3, our point feature renderer uses a modified U-Net architecture to generate colour outputs from the aggregated feature map. The architecture details are shown in Figure 10.

## A.3 Point Pruning

As described in Sec. 3.4, we use a background token $b$ to determine whether a ray intersects with the background. In our experiments, we set $b = 5$. Moreover, we modify the calculation of the attention weights $w_{i,j}$ for each point in Eqn. 6 to incorporate both the background token and the influence score, the updated definition is as follows:

$$w_{i,j} = \frac{w'_{i,j}}{\sum_{m=1}^{K} w'_{m,j}}, \text{ where } w'_{i,j} = \frac{\exp(a_{i,j} \cdot \tau_i)}{\exp(b) + \sum_{m=1}^{K} \exp(a_{m,j} \cdot \tau_m)} \tag{10}$$

Here, $a_{i,j}$ is defined in Eqn. 6.

Figure 11 shows a comparison between the point cloud learnt by the model with and without pruning on the Lego scene. The results demonstrate that the adopted pruning strategy yields a cleaner point cloud with fewer outliers.

## A.4 Point Growing

As mentioned in Sec. 3.4, our method involves growing points in the sparser regions of the point cloud. To identify these sparse regions, we compute the standard deviation $\sigma_i$ of the distances between

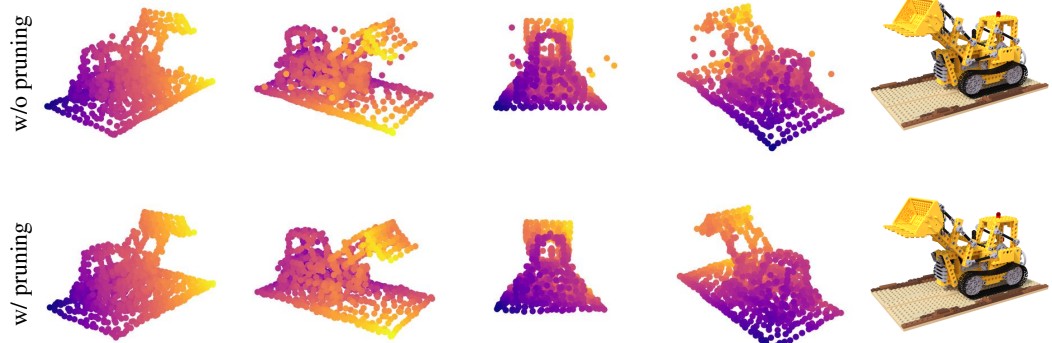

Figure 11: Comparison between point clouds learnt from the model without pruning (top row) and the model with pruning (bottom row). The point cloud learnt from the model with pruning exhibits cleaner structure and fewer outliers compared to the point cloud trained without pruning.

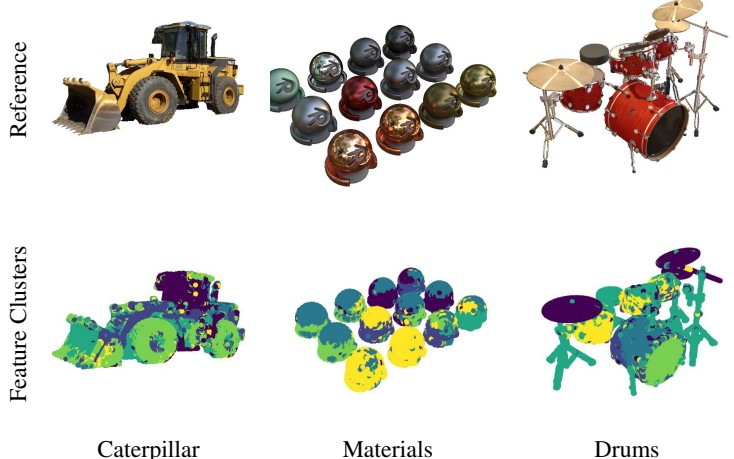

Caterpillar       Materials       Drums

Figure 12: Visualization of the clustered point feature vectors. In the bottom row for each scene, each colour represents a feature cluster. The clusters are obtained by clustering the learnt point feature vectors by the K-Means clustering algorithm with $K = 6$.

each point $i$ and its top 10 nearest neighbours. Subsequently, we insert new points near the points with higher values of $\sigma_i$. The location of the new point near point $i$ is selected as a random convex combination of $i$ and its three nearest neighbours.

## A.5   Clustering Feature Vectors

To demonstrate the effectiveness of our learnt point feature vectors, we visualize the feature vector clustering in Fig.12. The clustering is obtained by applying the K-Means clustering algorithm with $K = 6$ to the learnt feature vectors of the points. The resulting clustering reveals distinct separation, indicating the successful encoding of local information at surfaces with different textures and geometries.

More specifically, the feature vectors capture various colours within the scene. In the caterpillar scene, for instance, the point features associated with gray-coloured areas, such as the wheels and the side of the shovel, are clustered together. Moreover, the feature vectors can effectively differentiate between different materials, even when their colours may appear similar. For instance, in the materials scene, two balls on the bottom right most side have similar colours but different finishes—one with a matte appearance and the other with a polished look—and their feature vectors accurately distinguish between them. Additionally, the feature vectors demonstrate the capability to discern

variations in local geometry. In the drum scene, for example, the feature vectors for the large drum in the foreground differ from those representing the smaller drums in the background. These results highlight the encoding of distinct local scene information by the feature vectors, presenting potential utility in applications such as part segmentation.

### A.6 Zero-shot Geometry Editing and Object Manipulation

As mentioned in Sec.4.4, our model is capable of zero-shot geometry editing and object manipulation by modifying the scene point cloud. For zero-shot geometry editing, we achieve this by altering the positions of the points, while for object addition or removal, we duplicate or delete specific points accordingly. It is important to note that our approach eliminates the need for any additional pre- or post-processing steps for rendering the scene after the editing process. This shows the simplicity and robustness of our approach in editing the scenes.

### A.7 Texture Transfer

As mentioned in Sec. 4.4, to transfer textures between different parts of a scene, we first select the points representing the source and target parts, along with their corresponding point feature vectors. To capture the essential texture information, we identify the top $K$ principal components for both the source and target feature vectors. To facilitate texture transfer, we project the source feature vectors onto the target principal components and get the corresponding coordinates. Finally, we apply an inverse transformation to bring these coordinates back to the original space along the source principal components. To render an image with transferred textures during testing, we simply replace the texture vectors of the source points with the transferred texture vectors. The remaining components of the model remain unchanged during testing.

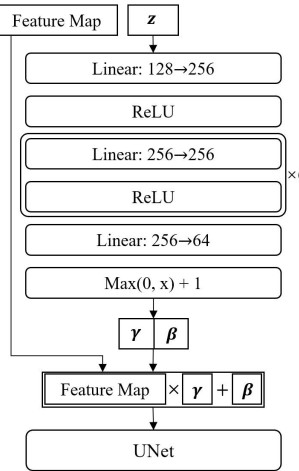

Figure 13: Modified point feature renderer architecture for exposure control, where we add an additional latent code input **z** which is fed into an MLP to produce affine transformation parameters for the feature map. The transformed feature map serves as the input to the U-Net architecture to produce the final RGB output image.

### A.8 Exposure Control

As exposure level is an uncontrolled variable in novel view synthesis of the scene, there can be multiple possible exposure levels for rendering unseen views not present in the training set. Therefore, we introduce an additional latent variable **z** to the model to explain this variability. Specifically, we modify the architecture of the point feature renderer $f_{\theta_{\mathcal{R}}}$ and use the latent code $\mathbf{z} \in \mathbb{R}^{128}$ as an additional input. The latent code is first passed through an MLP, which produces a scaling vector $\gamma \in \mathbb{R}^{32}$ and a translation vector $\beta \in \mathbb{R}^{32}$ for the feature map. The scaling vector $\gamma$ and the translation vector $\beta$ are applied channel-wise to the feature map. The output is then fed into the U-Net architecture (described in Sec. A.2) to produce the final RGB image. The architecture details are shown in Figure 13.

To effectively train a model to solve this one-to-many prediction problem, we employ a technique known as cIMLE (conditional Implicit Maximum Likelihood Estimation) [16]. cIMLE is specifically useful for addressing the issue of mode collapse and effectively capturing diverse modes within the target distribution. Algorithm 1 shows the pseudo code of cIMLE:

---

**Algorithm 1** Conditional IMLE Training Procedure

---

**Require:** The set of inputs $\{\mathbf{x}_i\}_{i=1}^n$ and the set of corresponding observed outputs $\{\mathbf{y}_i\}_{i=1}^n$
  Initialize the parameters $\theta$ of the generator $T_\theta$
  **for** $p = 1$ **to** $N_{outer}$ **do**
    Pick a random batch $S \subseteq \{1, \ldots, n\}$
    **for** $i \in S$ **do**
      Randomly generate i.i.d. $m$ latent codes
        $\mathbf{z}_1, \ldots, \mathbf{z}_m$
      $\tilde{\mathbf{y}}_{i,j} \leftarrow T_\theta(\mathbf{x}_i, \mathbf{z}_j) \ \forall j \in [m]$
      $\sigma(i) \leftarrow \arg\min_j d(\mathbf{y}_i, \tilde{\mathbf{y}}_{i,j}) \ \forall j \in [m]$
    **end for**
    **for** $q = 1$ **to** $N_{inner}$ **do**
      Pick a random mini-batch $\tilde{S} \subseteq S$
      $\theta \leftarrow \theta - \eta \nabla_\theta \left( \sum_{i \in \tilde{S}} d(\mathbf{y}_i, \tilde{\mathbf{y}}_{i,\sigma(i)}) \right) / |\widetilde{S}|$
    **end for**
  **end for**
  **return** $\theta$

---

In our specific context, $\mathbf{x}_i$ is the feature map, $T_\theta$ is the modified point feature renderer, $\mathbf{y}_i$ is the target RGB image. We first load pre-trained weights for all model parameters except the additional MLP for the latent variable before training. During test time, we randomly sample latent codes to change the exposure of the rendered image.

# B  Additional Results

## B.1  Qualitative Comparison

We include extra qualitative results for NeRF Synthetic dataset [22] in Figure 14. Furthermore, we include qualitative comparisons between our method and the more competitive baselines, based on metric scores, on the Tanks & Temples subset in Figure 15.

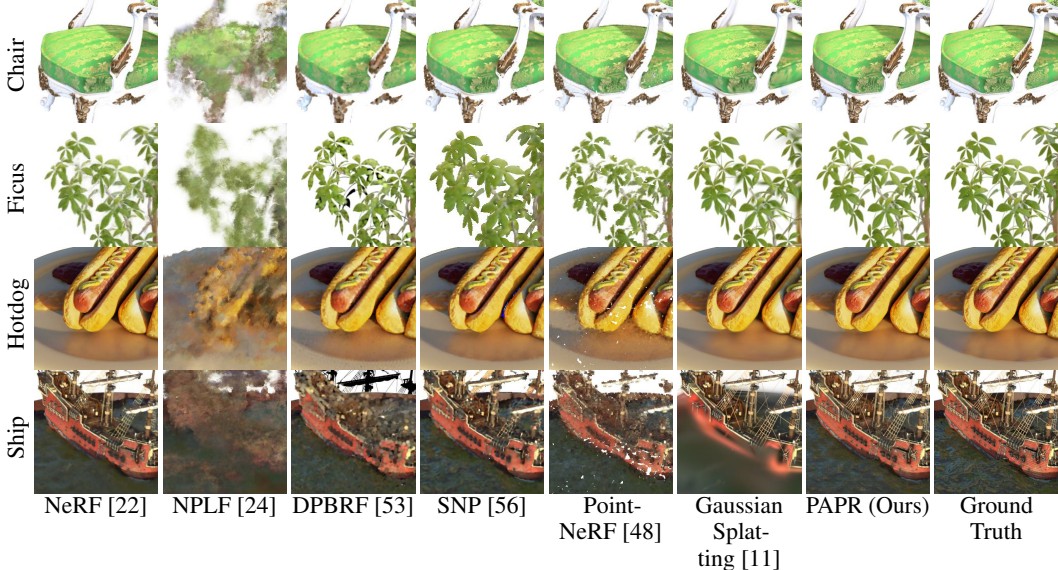

Figure 14: Qualitative comparison of novel view synthesis on the NeRF Synthetic dataset [22].

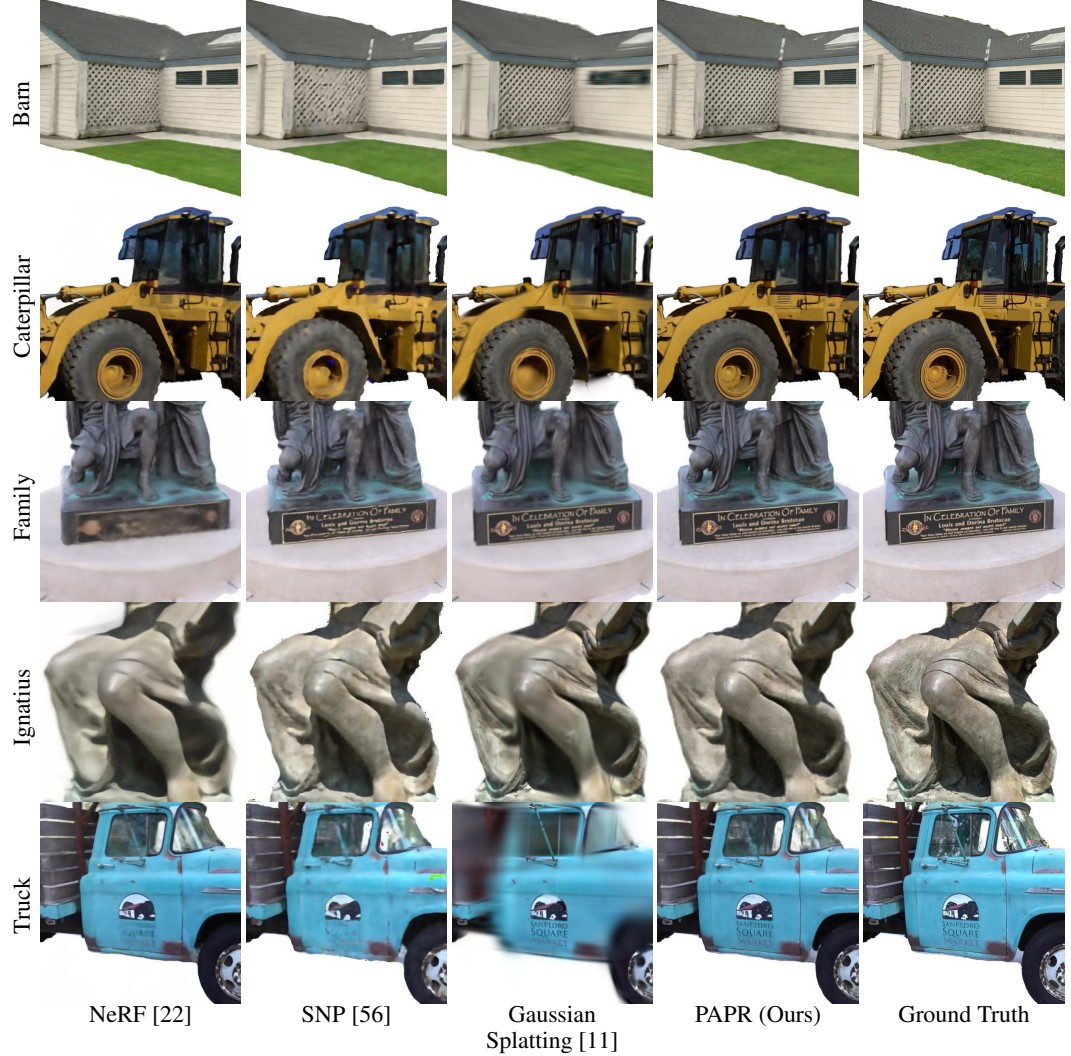

NeRF [22]     SNP [56]     Gaussian Splatting [11]     PAPR (Ours)     Ground Truth

Figure 15: Qualitative comparison of novel view synthesis between our method, PAPR, and the more competitive baselines on Tanks & Temples [13] subset.

## B.2 Point Cloud Learning

We show the point clouds generated by our method for each scene, depicted in Figure 16 for the NeRF Synthetic dataset [22], and Figure 17 for the Tanks & Temples [13] subset. These results demonstrate the effectiveness of our method in learning high-quality point clouds that correctly capture the intricate surfaces of the scenes.

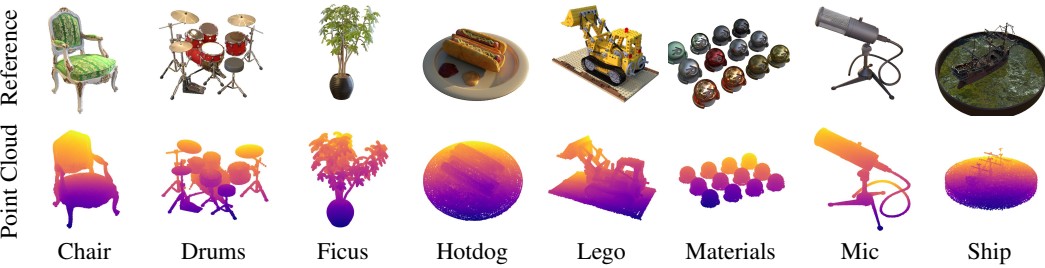

Chair    Drums    Ficus    Hotdog    Lego    Materials    Mic    Ship

Figure 16: Point clouds learnt by our method on the NeRF Synthetic dataset [22].

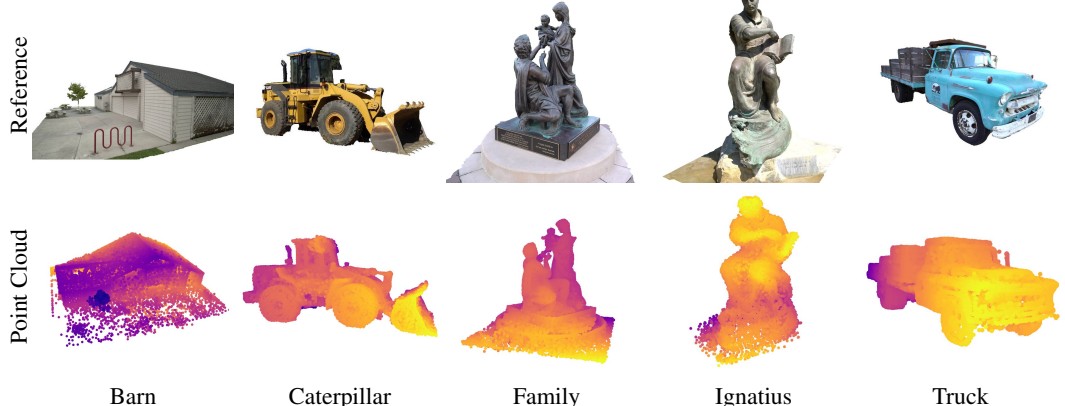

Figure 17: Point clouds learnt by our method on the Tanks & Temples [13] subset.

## B.3 Zero-shot Geometry Editing

We present comparisons with point-based baselines on two scenes in which we apply non-volume preserving stretching transformations. In the first scene, we stretch the back of the chair, and in the second case, we stretch the tip of the microphone. As shown in Figure 18, NPLF [24], DPBRF [53], Point-NeRF [48] and Gaussian Splatting [11] either create holes or produce significant noise after the transformation. Additionally, SNP [56] fails to preserve the texture details following the edits, such as the golden embroidery pattern on the back of the chair and the mesh grid pattern on the tip of the microphone. In contrast, our method successfully avoids creating holes and effectively preserves the texture details after the transformation.

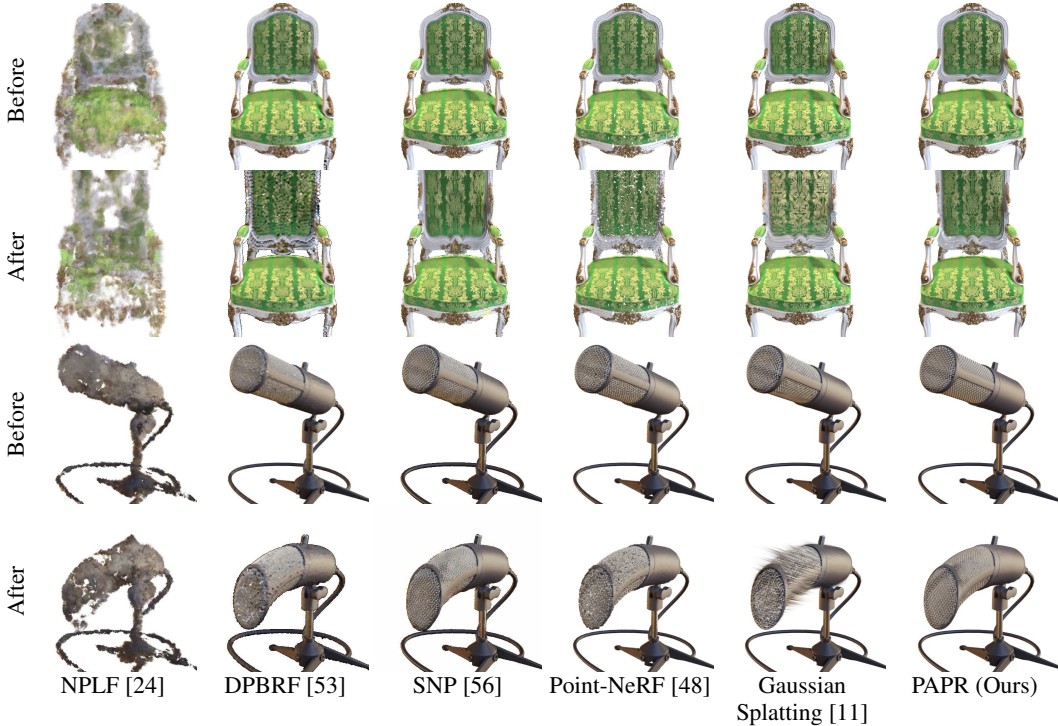

Figure 18: Qualitative comparisons among our method, PAPR, and point-based baselines following non-volume preserving stretching transformations. The rendered images before and after the transformations are shown. The results after the transformations are rendered by manipulating the point positions only during test time.

## B.4 Quantitative Results

We provide the metric scores broken down by scene on both datasets. Table 2 shows the per-scene scores for the NeRF Synthetic dataset [22] and Table 3 shows the scores for the Tanks & Temples [13] subset.

| | Chair | Drums | Lego | Mic | Materials | Ship | Hotdog | Ficus | Avg. |
|---|---|---|---|---|---|---|---|---|---|
| | | | | | PSNR↑ | | | | |
| NPLF [24] | 19.69 | 16.45 | 20.70 | 19.84 | 16.03 | 16.04 | 18.56 | 19.55 | 18.36 |
| DPBRF [53] | 26.51 | 19.38 | 25.09 | 29.26 | 26.20 | 21.93 | 31.87 | 24.61 | 25.61 |
| SNP [56] | 28.81 | 21.74 | 25.75 | 28.17 | 24.04 | 23.55 | 31.74 | 24.23 | 26.00 |
| Point-NeRF [48] | 30.24 | 23.60 | 23.42 | 31.75 | 24.77 | 18.72 | 26.56 | 28.40 | 25.93 |
| Gaussian Splatting [11] | 31.40 | 22.44 | 28.18 | 31.96 | 27.39 | 19.84 | 36.19 | 24.66 | 27.76 |
| PAPR(Ours) | **33.59** | **25.35** | **32.62** | **35.64** | **29.54** | **26.92** | **36.40** | **36.50** | **32.07** |
| | | | | | SSIM↑ | | | | |
| NPLF [24] | 0.798 | 0.737 | 0.763 | 0.886 | 0.739 | 0.678 | 0.792 | 0.842 | 0.780 |
| DPBRF [53] | 0.929 | 0.834 | 0.885 | 0.951 | 0.895 | 0.713 | 0.953 | 0.909 | 0.884 |
| SNP [56] | 0.938 | 0.889 | 0.907 | 0.964 | 0.909 | 0.823 | 0.961 | 0.917 | 0.914 |
| Point-NeRF [48] | 0.971 | 0.927 | 0.904 | 0.985 | 0.929 | 0.761 | 0.934 | 0.973 | 0.923 |
| Gaussian Splatting [11] | 0.957 | 0.901 | 0.935 | 0.982 | 0.938 | 0.792 | 0.978 | 0.953 | 0.929 |
| PAPR(Ours) | **0.986** | **0.951** | **0.981** | **0.993** | **0.972** | **0.904** | **0.988** | **0.994** | **0.971** |
| | | | | | LPIPS$_{Vgg}$↓ | | | | |
| NPLF [24] | 0.183 | 0.214 | 0.236 | 0.136 | 0.226 | 0.339 | 0.226 | 0.143 | 0.213 |
| DPBRF [53] | 0.098 | 0.174 | 0.142 | 0.070 | 0.143 | 0.288 | 0.093 | 0.095 | 0.138 |
| SNP [56] | 0.049 | 0.081 | 0.057 | 0.025 | 0.072 | 0.167 | 0.036 | 0.050 | 0.110 |
| Point-NeRF [48] | 0.065 | 0.125 | 0.152 | 0.038 | 0.148 | 0.298 | 0.143 | 0.067 | 0.129 |
| Gaussian Splatting [11] | 0.049 | 0.108 | 0.077 | 0.019 | 0.063 | 0.273 | 0.037 | 0.049 | 0.084 |
| PAPR(Ours) | **0.018** | **0.055** | **0.027** | **0.007** | **0.036** | **0.129** | **0.021** | **0.010** | **0.038** |

Table 2: Comparison of image quality metrics (PSNR, SSIM and LPIPS [54]), broken down by scene, for the NeRF Synthetic dataset [22].

## B.5 Ablation Study

We provide additional quantitative results for the ablation study. Figure 19 shows the PSNR and SSIM scores for the methods using different number of points. Additionally, Table 4 shows the PSNR and SSIM scores for the various choices of the ray-dependent point embedding. These results validate the effectiveness of our proposed method.

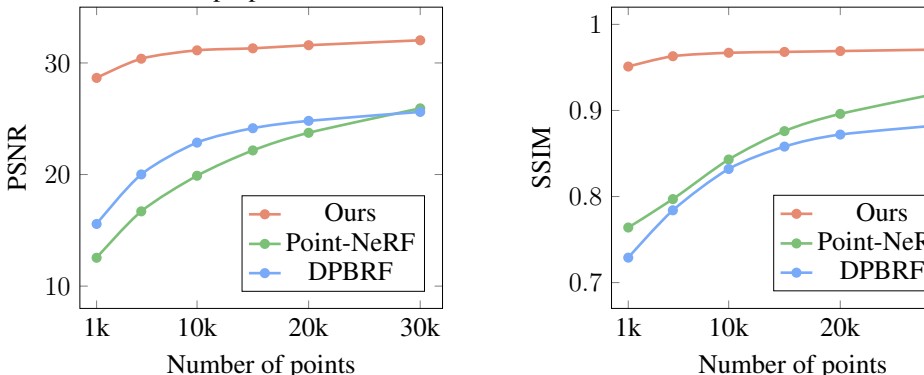

Figure 19: Extra image quality metrics (PSNR, SSIM) for the ablation study on different number of points. Higher values of PSNR and SSIM scores are better.

|  | Ignatius | Truck | Barn | Caterpillar | Family | Avg. |
|---|---|---|---|---|---|---|
| | | | PSNR↑ | | | |
| NPLF [24] | 24.60 | 20.01 | 20.24 | 17.69 | 23.39 | 21.19 |
| DPBRF [53] | 18.22 | 16.20 | 14.68 | 17.23 | 19.93 | 17.25 |
| SNP [56] | 28.41 | 24.32 | 25.27 | 22.62 | 31.31 | 26.39 |
| Point-NeRF [48] | **28.59** | 25.57 | 20.97 | 17.71 | 30.89 | 24.75 |
| Gaussian Splatting [11] | 26.48 | 22.32 | 26.22 | 23.27 | **35.79** | 26.81 |
| PAPR(Ours) | 28.40 | **26.98** | **27.06** | **26.79** | 34.39 | **28.72** |
| | | | SSIM↑ | | | |
| NPLF [24] | 0.882 | 0.725 | 0.634 | 0.719 | 0.846 | 0.761 |
| DPBRF [53] | 0.797 | 0.563 | 0.422 | 0.614 | 0.776 | 0.634 |
| SNP [56] | 0.944 | 0.876 | 0.832 | 0.858 | 0.958 | 0.894 |
| Point-NeRF [48] | **0.960** | 0.926 | 0.799 | 0.797 | 0.970 | 0.890 |
| Gaussian Splatting [11] | 0.937 | 0.867 | 0.860 | 0.889 | **0.983** | 0.907 |
| PAPR(Ours) | 0.956 | **0.931** | **0.896** | **0.932** | **0.983** | **0.940** |
| | | | LPIPS$_{Vgg}$↓ | | | |
| NPLF [24] | 0.120 | 0.265 | 0.381 | 0.275 | 0.157 | 0.240 |
| DPBRF [53] | 0.174 | 0.317 | 0.484 | 0.334 | 0.194 | 0.301 |
| SNP [56] | 0.077 | 0.177 | 0.278 | 0.196 | 0.071 | 0.160 |
| Point-NeRF [48] | 0.085 | 0.162 | 0.355 | 0.237 | 0.084 | 0.184 |
| Gaussian Splatting [11] | 0.095 | 0.186 | 0.231 | 0.155 | 0.032 | 0.140 |
| PAPR(Ours) | **0.072** | **0.108** | **0.157** | **0.118** | **0.031** | **0.097** |

Table 3: Comparison of image quality metrics (PSNR, SSIM and LPIPS [54]), broken down by scene, for the Tanks & Temples [13] subset.

|  | Lego | | Mic | |
|---|---|---|---|---|
| | *PSNR* ↑ | *SSIM* ↑ | *PSNR* ↑ | *SSIM* ↑ |
| Full Model | **32.62** | **0.981** | **35.64** | **0.993** |
| w/o $p_i$ | 31.67 | 0.976 | 32.00 | 0.985 |
| w/o $p_i, t_{i,j}$ | 13.26 | 0.701 | 18.73 | 0.902 |

Table 4: Extra image quality metrics (PSNR, SSIM) for the ablation study on different designs for the ray-dependent point embedding. Higher values of PSNR and SSIM scores are better.

