# OpenReview forum: "PAPR: Proximity Attention Point Rendering"
_NeurIPS.cc/2023/Conference — NeurIPS 2023 spotlight_

### Official Review · Reviewer_U5yB · 2023-07-02

**Soundness:** 3 good
**Presentation:** 3 good
**Contribution:** 3 good
**Rating:** 4
**Confidence:** 4

**Summary:**

This paper proposes a neural point renderer that improves the existing differentiable point rasterizers using a learned attention block, which improves the gradient behavior alleviating the "vanishing gradient" issue that comes with many existing differentiable point rasterizer. The method compares favorably with prior point-based NeRFs, jointly optimizing the point features and point locations, and achieving impressive 2D visual and 3D reconstruction quality even when using only poorly initialized sparse points. Additionally, it demonstrates useful applications thanks to the explicit point representation.

**Strengths:**

- The proposed method uses a learned attention block to determine the blending weights. It engineers the query, value and key embeddings using suitable geometric entities, which seems oddly effective (see my questions below) to emulate the similar effect of a fully deterministic point rasterizer in terms of the blending behavior w.r.t. point relative distance and depth.
- The learning is further assisted with progressive point growing and pruning, which, as demonstrated in the ablation experiments, are effective techniques to achieve high quality reconstruction.
- The visualization of point features through clustering is very interesting and insightful.
- The paper is well-written, carefully explained, and the proposed components are thoroughly evaluated.
- The improvement over existing point-based NeRF papers are significant.

**Weaknesses:**

1. No discussion about complexity. This is important, since the proposed method boasts to be able to compute gradient for all points regardless of their absolution distance to the target pixel in the pixel-space. This requires computing the attention with *every point* for every ray. Existing methods using RBF or other definition rasterization gradient (e.g. [44]) utilize handcrafted rules to reduce the impact of a pixel to a point as the point-pixel increases. While these rules may have the vanishing gradient problem the authors identified, they can serve as hard constraint to limit the number of target pixels required to evaluate the gradient for a specific point, i.e. pixels outside a search window can be ignored. With the proposed approach, and the claimed benefit, that each point receives gradient from all pixels, the computation is significantly higher. It's therefore important to breakdown the algorithm complexity of each step.

1 (cont). Besides the theoretical complexity analysis, the authors should also provide the training and inference speed compared to a standard NeRF that achieves similar visual quality.

2. the remark to [44] is not correct. [44] *is* able to render colored and shaded points. Besides, SynSin [Wiles et.al. CVPR 2020], Pulsar [Lassner, Christoph, and Michael Zollhofer. CVPR 2021] and Adop [35] are three relevant papers that should be cited too. In addition, I would argue that [44], Pulsar and [35] are more relevant than the baselines that are based on MVS initialization, and should be compared too.

3. The necessity to use a convolutional U-Net in the end is disappointing, as the convolution is likely to induce flickering and other inconsistencies when moving the camera, which could be the cause of the flickering on drum surface in the video?


**Questions:**

I'd like the authors to address my concerns in the "weakness" section. Besides, please also help me understand the following questions in the rebuttal.

1. Since the relative distance is completely learned without any prior, why would the network learn to produce higher weights for closer points. Could you visualize the learned weights for some sampled rays? Please also clarify whether Fig 3c is computed from a real learned result or plotted as the "desired behavior"? In addition, why do the curves go up again when the points move further away from the pixel (the right-most part of the blue curve and the left-most part of the orange curve).

1. (cont) The inputs to the key and query embeddings don't share any similarity at all. The MLPs need to learn to map them to compatible space and ensure that keys (3) of points that are close to the rays (which would correspond to the Fig 3c gradient behavior) are more similar to the query (5). How does the network learn this without any prior? Do you need special initialization for the MLPs? And what is the convergence rate.

2. Ideally the learnable foreground score should be related depth and occlusion. But without constraint, this learned foreground score can be abused to label "outlier points" that are far from any target pixels due to optimization problems, which can be conveniently pruned and therefore ignored in the optimization. Please visualize the foreground score before point pruning to ensure that this score indeed correspond to visibility.



**Limitations:**

Yes.

---

> ### Author Rebuttal · Authors · 2023-08-10
>
> ### Q1: Computation complexity of attention per ray.
>
> A1: Actually we don't compute the attention with every point for each ray. As mentioned on L164-166, instead of attending to all points, we attend to the top K nearest points around each ray. This “K” serves as a hard constraint for the computation cost for each ray. In our experiments, we choose K=20 which is a lot smaller than the overall point count of 30k.
>
> ### Q2: Training and inference speed?
>
> A2: To render an 800x800 image, our model takes 2.8s, while NeRF would take 11.8s. Our code's inference speed is not yet optimized, with the standard PyTorch topk function constituting about 96% of total inference runtime. Utilizing custom CUDA kernels could potentially accelerate inference speed further.
>
> In terms of training time, our method takes about 20 hours to train a model for a scene from either NeRF Synthetic or Tanks & Temples datasets. In comparison, NeRF would require 20 hours to train on a scene from the NeRF synthetic dataset, and 30 hours on a scene from Tanks & Temples subset.
>
> ### Q3: Remark on [44].
>
> A3: Thank you for the correction - we will revise in the camera-ready.
>
> ### Q4: SynSin [Wiles et al.], Pulsar [Lassner et al.] and Adop [35] should be cited too.
>
> A4: In fact, these works have been cited already, as [45], [15] and [35].
>
> ### Q5: Comparison to [44], [15] and [35].
>
> A5: Given the limited timeframe of the rebuttal period, we managed to run two of the more recent baselines, Pulsar [15] and Adop [35], and include the results of the comparison in the rebuttal PDF. As shown, our method outperforms the baselines and contains more details in the rendering.
>
> The DSS [44] codebase is older and so it is no longer compatible with later versions of CUDA that are required by our GPU hardware. Despite investing substantial time and effort, we did not manage to upgrade their codebase to be compatible with our current hardware setup.
>
> ### Q6: The necessity to use a convolutional U-Net in the end is disappointing, as the convolution is likely to induce flickering and other inconsistencies when moving the camera.
>
> A6: Sure, the use of convolutional U-Net may cause flickering. However, it's important to note that using a convolutional U-Net isn't mandatory for the renderer. In fact, we present an ablation study below, wherein the U-Net is replaced by an MLP on the NeRF Synthetic dataset.
>
> | | PSNR | SSIM | LPIPS-vgg |
> | --- | --- | --- | --- |
> | MLP | 30.54 | 0.962 | 0.054 |
> | U-Net  | 32.07 | 0.971 | 0.038 |
>
> The table above demonstrates that even with the substitution of an MLP for the renderer, our model maintains competitive performance. Importantly, our primary contribution doesn't hinge on the specific choices of renderer architecture.
>
> ### Q7: Since the relative distance is completely learned without any prior, why would the network learn to produce higher weights for closer points. Could you visualize the learned weights for some sampled rays?
>
> A7: Certainly. The visualization of the attention weights of the points around a given ray is shown in the rebuttal PDF. As shown, the model does indeed assign higher weights to closer points. Intuitively, if the attention mechanism assigns high weights for points that are far away, it would be difficult to preserve the 3D consistency. For instance, when two rays with different camera centers both shoot rays towards the same point in the foreground scene, the rendered colour should be very similar. However, when the attention favours points far away, there could be many far away points for each ray to choose from, and it’s easy to choose different points. This would result in rendering a different colour for each different view for the same 3D location. Therefore, the best way to preserve consistency in rendered colour is by assigning high attention weight to points close to the ray.
>
> ### Q8: Please also clarify whether Fig 3c is computed from a real learned result or plotted as the "desired behavior"?
>
> A8: Fig 3c is the “desired behaviour”.
>
> ### Q9: Why do the curves go up again when the points move further away from the pixel?
>
> A9: As the pixel moves to the right away from i, it first gets close to j and then moves past j. As it approaches j, it gets a lot closer to j than i, and so the ratio of its distance to i to the sum of its distances to i and j gets large, and so the scale of the gradient w.r.t. i gets small. However, as it moves past j and gets far from it, the ratio of its distance to i to the sum of its distances to i and j eventually decreases to 1/2, and so the the scale of the gradient w.r.t. i gets large again.
>
> ### Q10: How does the network learn to map the key and query to similar space without any prior?
>
> A10: Similar to what is explained earlier, the network is encouraged to assign high weights to closer points to better preserve 3D consistency. Therefore, the MLPs would learn a feature space where the proximity between the point embedding and the ray corresponds to the proximity between their respective features, thereby allowing closer points to receive higher weights.
>
> ### Q11: Do you need special initialization for the MLPs?
>
> A11: No. The weights to the MLPs are randomly initialized.
>
> ### Q12: What is the convergence rate?
>
> A12: We have include a loss curve throughout the course of training in the rebuttal PDF. As shown, our model converges smoothly and quickly.
>
> ### Q13: Please visualize the foreground score.
>
> A13: Actually the learnable foreground score is not necessarily related to depth and occlusion, because we assume a constant coloured background, which does not need to be modelled with points. Hence, all points are visible from some views. Instead it is used to upweight the contributions of particular points, and so is more related to the certainty in the model’s belief in the point being in the foreground and its opaqueness. We include a visualization of the foreground scores before pruning in the rebuttal PDF. As shown, they behave as intended.

---

> > ### Author Response · Authors · 2023-08-21
> > **Discussion Ending Soon (less than 14 hours) : Would appreciate any feedback from Reviewer U5yB**
> >
> > Since there are less than 14 hours remaining in the discussion period (deadline is on Aug 21st at 1 pm EDT) and we haven’t received your response for our rebuttal, we thought we’d reach out to you directly. We hope our response addressed your concerns. Your input is valuable to us, and we would be happy to address any remaining concerns if you still have any. If all your concerns have been addressed, we would be grateful if you would kindly consider updating your rating. Thank you again for taking the time and effort to review our paper.

---

> > > ### Comment · Area_Chair_3N4b · 2023-08-22
> > >
> > > Dear authors, ``U5yB`` has commented on your points in a post that was not visible to you. The AC will take these into consideration when making a recommendation.

---

### Official Review · Reviewer_TJqY · 2023-07-06

**Soundness:** 4 excellent
**Presentation:** 4 excellent
**Contribution:** 4 excellent
**Rating:** 8
**Confidence:** 4

**Summary:**

This paper presents a novel view synthesis and 3D reconstruction framework via a point-based representation and a jointly learned transformer-based differentiable renderer. Given multi-view images of a foreground-centric scene, the method can effectively deform a point-cloud from scratch to match the surfaces, with an optional pruning and growing mechanism. Each point is associated with a learned feature, which allows novel views of the scene to be rendered through a transformer-based point renderer.

Experiments show that this method leads to high-quality geometry and novel view synthesis results, significantly outperforming existing point-based NVS methods as well as the vanilla NeRF.

**Strengths:**

### S1 - A solid pipeline for point-based reconstruction and rendering
- The proposed pipeline is very well designed and validated. In particular:
    - The design of the transformer-based point renderer is interesting. Ray direction is treated as the query feature, point position the key, and point features the value, which make a lot of sense.
    - The transformer directly outputs the weights of each point for integration, without a separate volume rendering function as in Point-NeRF that would introduce another order of computation.

### S2 - Great results on geometry reconstruction and novel view synthesis
- The method effectively deforms a point cloud from a initial sphere to complex surfaces, which is a very compact representation of the scene geometry, and allows for various scene editing.
- The NVS results are also quite impressive, even outperforming NeRF (although I guess one might be able to further tune NeRF to achieve even better numbers).

### S3 - Good demonstrations of scene editing with points
- The paper presents a number of scene editing applications, including deformation, object removal and appearance editing. Having an explicit point-based representation makes such editing much easier than implicit NeRF/volumes.

### S4 - Good presentation
- The paper is well written, with clear motivation and illustrative figures.

**Weaknesses:**

### W1 - Renderer seems to be learned jointly on one scene only
- In theory, the transformer-based renderer should be generic and can be shared across any scene?
Does it generalize to other scenes?

### (minor) W2 - Technical details
- There seems no explicit inductive bias in the renderer that would encourage it to actually put higher weights to points closer to the ray and to the camera (within the K nearest points). Does the learned renderer actually utilizes the information of $\mathbf{s}_{i,j}$ and $\mathbf{t}$ as we expect? It would be helpful to visualize the weights.
- Sup. mat. Sec. B.3 Eq (10): Is the $w_{i,j}$ here taken before or after the softmax in Eq. (6)?
    - If before, $w_{i,j}$ is not guaranteed to be positive, and Eq. (7) and Eq. (10) would be problematic, as both $w_{i,j}$ and $\tau_i$ can be either positive or negative.
    - If after, there would be double softmax / exponential. Does that cause any issue, eg. vanishing gradients?
- Pruning points with $\tau_i \lt 0$ seems quite arbitrary. I suppose this is related to the choice of the predefined background weight scalar $b$. Is the threshold $0$ chosen empirically? (why not, say, $\tau_i \lt 1$ or $\tau_i \lt -2.64$?)
- What is the run time? How does it compare to say NeRF?
- What happens if it applied to forward facing scenes with complex background, eg. LLFF? The limitation discusses the failure in modeling background in 360° scenes, but what about forward-facing scenes, where points can be legitimately assigned to "background"?

### Additional comments
- Fig. 4: When the pruning is disabled, it is not surprising at all that Point-NeRF fails to represent the surface using the points, as it is still using a volume rendering function where the underlying geometry is ultimately still represented by the density values. Although, I agree it is great that the proposed method is able to deform the points correctly around the surface without pruning, thanks to the distance-aware weighting in the rendering function I suppose.
- Only one attention layer?
- Any points remain inside the object?
- Line 109: minor grammatical issues.

**Questions:**

No critical concerns. It would be helpful to elaborate on the generalization of the jointly learned transformer renderer.

**Limitations:**

The paper includes a nice discussion on the limitation regarding background modeling.

---

> ### Author Rebuttal · Authors · 2023-08-10
>
> ### Q1: In theory, the transformer-based renderer should be generic and can be shared across any scene? Does it generalize to other scenes?
>
> A1: Agreed, the architecture should generalize and it would be an interesting direction to explore in future work.
>
> ### Q2: Does the learned renderer actually utilizes the information of $s_{i, j}$ and $t$ as we expect?
>
> A2: We have included a visualization of the attention weights of the points for a given ray in the rebuttal PDF. As shown in the figure, the attention mechanism does place higher weights on points closer to the ray.
>
> ### Q3: Sup. mat. Sec. B.3 Eq (10): Is the $w_{i, j}$ here taken before or after the softmax in Eq. (6)?
>
> A3: Good catch. There was a typographical error in the equation. The $w_{i, j}$ should indeed be before the softmax, and should have had a ReLU applied to ensure non-negativity. Concretely, the softmax in Eq. (6) should have been moved outside of the definition of $w_{i, j}$, and ReLUs should be applied on $w_{i, j}$ in Eqs. (7) and (10). So the correct formulas for Eqs. (6), (7) and (10) are:
>
> $w_{i,j} = \frac{<\mathbf{q}_{j}, \mathbf{k}_{i,j}>}{\sqrt{\mu}}$
>
> $w_{b,j} = \frac{\text{exp}(b)}{\text{exp}(b) + \sum^K_{m=1}\text{exp}(\max(0, w_{m,j})\cdot \tau_m)}$
>
> $a_{i,j} = \frac{\text{exp}(\max(0, w_{i,j})\cdot \tau_i)}{\text{exp}(b) + \sum^K_{m=1}\text{exp}(\max(0, w_{m,j})\cdot \tau_m)}$
>
> We will update the formulas in the camera-ready.
>
> ### Q4: I suppose the pruning threshold is related to the choice of the predefined background weight scalar $b$. Is the threshold for $\tau_i$ chosen empirically?
>
> A4: Actually the pruning threshold is not related to the choice of $b$. It’s chosen to be 0 based on the behaviour of the attention weights as a function of $\tau_i$. Specifically, if $\tau_i$ is negative, then the higher $\max(0, w_{m,j})$ gets, the lower the attention weight becomes. Intuitively, this means that the point is a distractor and so should be pruned.
>
> ### Q5: What is the run time? How does it compare to say NeRF?
>
> A5: To render an 800x800 image, our model takes 2.8s, while NeRF would take 11.8s. Our code's inference speed is not yet optimized, with the standard PyTorch topk function constituting about 96% of total inference runtime. Utilizing custom CUDA kernels could potentially accelerate inference speed further.
>
> ### Q6: What happens if it applied to forward facing scenes with complex background, eg. LLFF, where points can be legitimately assigned to "background”?
>
> A6: This depends on whether the background is given. If not, then the method would would face the same issue as the 360-degree scenes.
>
> ### Q7: Only one attention layer?
>
> A7: Yes.
>
> ### Q8: Any points remain inside the object?
>
> A8: While there may be some points inside, the vast majority are on the surface.
>
> ### Q9: Line 109: minor grammatical issues.
>
> A9: Thank you for pointing it out, we will revise it in the camera-ready.

---

> > ### Comment · Reviewer_TJqY · 2023-08-16
> > **Thanks for the detailed responses**
> >
> > I appreciate the immense effort the authors put in the rebuttal to provide the additional results and elaborate technical explanations. In particular, the visualization of the weight distributions and the ablation studies on various token design are very helpful.
> >
> > I think this is a very solid implementation for a learning-based point renderer and my rating remains. I would very much look forward to a version of this renderer being trained at a larger scale that can generalize across arbitrary scenes.

---

> > > ### Author Response · Authors · 2023-08-17
> > > **Thank you for the response**
> > >
> > > Thank you for acknowledging our efforts in the rebuttal, and for recognizing the quality of our work! We're pleased that the weight distribution visualizations and ablation studies were helpful. We are also excited about extending the current method to allow generalization across scenes, and will certainly take your suggestions into account in our future work.

---

### Official Review · Reviewer_dttq · 2023-07-07

**Soundness:** 3 good
**Presentation:** 3 good
**Contribution:** 3 good
**Rating:** 6
**Confidence:** 3

**Summary:**

This paper proposes a novel method that consists of a point-based scene representation and a differentiable renderer. The proposed representation utilizes relative distance to characterize the contribution of each point, preventing the vanishing gradient issue. The method outperforms baselines in terms of optimizing point cloud positions, rendering. And it enables interesting applications.

**Strengths:**

- Overview figure is clear, well-conveying the idea of the paper.

- The representation is simple, reasonable and well-desinged.

- The proposed method outperforms the baselines.

**Weaknesses:**

The paper is sound in general. However there are several points that can be clarified.

- Feature vector: learned or deterministically determined? If it is the latter case, how it can be determined?

- In the figure 3-(c), why does the gradient scale increases again when the target pixel position deviates much from the point cloud?

- In L148-151, it is mentioned that using the relative distance guarantess that there are always some points contributing to each pixel, even if their absolute distances are large.
However, if the distance is too far, wouldn't this design enlarge the error, i.e., generating the color of the unrelated point cloud? I think setting a threshold would be helpful to prevent error, or is this thresholding same as the filtering process in L164?. Also, if the filtering is applied (L164), wouldn't the points that are far away be removed and therefore cannot contribute to the pixel? And how is this filtering threshold decided? Would the value of it affect the performance?

- How can the view dependent effect be modeled? From my understanding, different viewing direction results in different s,t value of each point, and thus it is possible to model it?

- What is the fixed background token b in L189?

- In Figure 5, the result of proposed method on the Mic sample shows very fine-grained detail similar to GT. However, the result on the Materials sample is relatively blurry.
What is the reason behind the quality difference?

- It would be great if the visual ablations can be done when using the different number of points.


**Questions:**

- How can the transparent material modeled with the proximity attention?

- How this approach can be extended to complicated scene with lot of occlusions?

- This approach can work on rigid deformation for some extent. How can this approach be extended to non-rigidly deforming scene?

**Limitations:**

- Clarifying some details (as mentioned in the weakness section) could further improve the paper.

---

> ### Author Rebuttal · Authors · 2023-08-10
>
> ### Q1: Is feature vector learned or deterministically determined?
>
> A1: The feature vector is learnt as mentioned in L207 - 208.
>
> ### Q2: In the figure 3-(c), why does the gradient scale increases again when the target pixel position deviates much from the point cloud?
>
> A2: As the pixel moves to the right away from i, it first gets close to j and then moves past j. As it approaches j, it gets a lot closer to j than i, and so the ratio of its distance to i to the sum of its distances to i and j gets large, and so the scale of the gradient w.r.t. i gets small. However, as it moves past j and gets far from it, the ratio of its distance to i to the sum of its distances to i and j eventually decreases to 1/2, and so the the scale of the gradient w.r.t. i gets large again.
>
> ### Q3: If the distance is too far, wouldn't this design (using relative distance) enlarge the error, i.e., generating the color of the unrelated point cloud?
>
> A3: No. The “error” mentioned here is actually the supervision signal, which is essential for learning from scratch. In cases where the model produces a colour at a distant pixel that has a high loss value, the training process would update the point positions and feature vectors to produce a colour that better matches the ground truth, consequently reducing the error. Therefore, when the training has converged, the model would have learnt to produce the correct colour for every pixel, including the ones that are distant from the points.
>
> ### Q4: I think setting a threshold would be helpful to prevent error, or is this thresholding same as the filtering process in L164?
>
> A4: No, the filtering process in L164 is different from thresholding. To illustrate the difference, imagine a scene where one region contains points lying on a sparse grid, while an adjacent region contains points on a dense grid. Now, let's examine the rays intersecting these two regions.
>
> If we were to apply a single threshold to rays at both regions, there would be more points that fall within the threshold in the dense region than the sparse region. However, by applying the filtering process in L164, we can ensure that there would be exactly K points selected near each ray, regardless of the region's density. This is unattainable through thresholding.
>
> ### Q5: Also, if the filtering is applied (L164), wouldn't the points that are far away be removed and therefore cannot contribute to the pixel? And how is this filtering threshold decided? Would the value of it affect the performance?
>
> A5: No, the filtering in L164 does not necessarily result in the removal of points that are far away. Instead, this process retains the top K nearest points around the ray, irrespective of their absolute distances from the ray. i.e., this approach ensures that even in cases where all points are far away from the pixel, the top K nearest points can still contribute to it.
>
> Regarding the impact of K on performance, using an excessively small K value could restrict the contextual information available to the attention mechanism, potentially leading to worse performance. In practice, we choose value of K=20 that is big enough to provide sufficient context to the model while maintaining a reasonable computational load.
>
> ### Q6: How can the view dependent effect be modeled? From my understanding, different viewing direction results in different s,t value of each point, and thus it is possible to model it?
>
> A6: Yes, that’s right.
>
> ### Q7: What is the fixed background token b in L189?
>
> A7: The background token enables the attention mechanism to handle background rays by assigning low weights to all points. Without this term “b”, the attention mechanism would not be able to do so. In our experiments, we set b=5, as specified in Section B.3 of the supplementary material.
>
> ### Q8: What is the reason behind the quality difference between the mic and the material sample in Figure 5?
>
> A8: The quality difference is because the Mic scene only has a single object with all of the points on its surface, whereas the material scene needs to distribute the same total number of points among twelve balls, resulting in less detail per ball.
>
> ### Q9: It would be great if the visual ablations can be done when using the different number of points.
>
> A9: Please refer to the rebuttal PDF for the qualitative results of the ablation on different number of points.
>
> ### Q10: How can the transparent material modeled with the proximity attention?
>
> A10: Fully transparent material can be modelled simply with an absence of points. If you mean translucent material, they may conceivably be modelled with a lower foreground score, which would cause a shift in the attention weights towards other points that are more opaque.
>
> ### Q11: How this approach can be extended to complicated scene with lot of occlusions?
>
> A11: This could be achieved by increasing both the total number of points and the hyperparameter K, which controls the filtering for the top K nearest point around each ray.
>
> ### Q12: How can this approach be extended to non-rigidly deforming scene?
>
> A12: Yes. In fact, we’ve showcased a non-rigid stretching deformation of the mic in the third column of Figure 8a.

---

> > ### Comment · Reviewer_dttq · 2023-08-18
> > **Response to the author rebuttal**
> >
> > I would like to thank the authors for their time and efforts dedicated to rebuttal. All of my concerns have been addressed.

---

> > > ### Author Response · Authors · 2023-08-19
> > > **Thank you for the response**
> > >
> > > Thank you for your feedback and recognition of our efforts. We're pleased to hear that our response addressed your concerns. If you have any more questions, please don't hesitate to let us know.

---

### Official Review · Reviewer_fb1M · 2023-07-07

**Soundness:** 4 excellent
**Presentation:** 4 excellent
**Contribution:** 3 good
**Rating:** 8
**Confidence:** 5

**Summary:**

This paper presents a novel point-based rendering approach, PAPR, to learn a point cloud for neural rendering under the supervision of posed RGB images. At the core of softening the gradient flow by using an attention mechanism in a novel and elegant way, the proposed PAPR enables high-fidelity rendering while explicitly learning the parsimonious point-based scene representation from scratch. Based on this, the proposed PAPR enjoys flexibility for geometry editing, object manipulation, texture transfer, and exposure control, all with impressive results.

**Strengths:**

1. Learning a parsimonious point cloud from scratch in the form of neural rendering without using the 3D scene information is very interesting and has many potential applications.

2. The core module, proximity attention, is well-designed and should be computationally efficient. Instead of sampling points along rays, the proximity attention module proposed in this paper uses the projection relationship between rays and points to aggregate the proximate points in an attention form and feed the point features in a U-Net for rendering. The overall pipeline is neat and technically sound. The computation should be efficient as well.


**Weaknesses:**

1. Because the authors emphasize the parsimony property of the point cloud, the number of points should be the key to be studied. Although the authors have provided an ablation study in Fig. 6 on the effect of the number of points on rendered image quality, some in-depth discussion would further strengthen this paper. Because the authors mentioned that the proposed method is currently suitable for the near-constant background color, I am curious about the parsimony of the point cloud concerning the non-constant background color.

2. The growing scheme is somewhat heuristic with predefined hyperparameters such as 500 iterations, top-10 nearest neighbors, etc. There should be some ablation study for the effect of these hyperparameters. Furthermore, because proximity attention could capture the relationship between points, is there a way to use the learned affinity to identify the sparse regions?



**Questions:**

Because this paper has shown their ability to learn a parsimonious point cloud from scratch, what will be if we have some inductive initialization of the point cloud? For example, could the proposed method refine the sparse reconstruction quality if we have a sparse reconstruction result for the scene by COLMAP? I am also curious about the potential of doing bundle adjustment with the proposed PAPR.


**Limitations:**

The authors have discussed their limitations. I don't have any concern on this point.

---

> ### Author Rebuttal · Authors · 2023-08-10
>
> ### Q1: The parsimony of the point cloud concerning the non-constant background color.
>
> A1: The parsimony of the point cloud is not related to the presence of a non-constant background colour, as we use the points to model the foreground scenes exclusively. To handle non-constant background, one idea is to learn a dedicated background model (e.g., a light field-based or NeRF-based model). The key is to separate the modelling of the foreground from the background so that the parsimony of the foreground representation is preserved.
>
> ### Q2: Ablation on the effects of point growing hyperparameters.
>
> A2: We show the results of different point growing interval  on the NeRF Synthetic dataset in the table below.
>
> | Point Growing Interval (# of iterations) | PSNR $\uparrow$ | SSIM $\uparrow$ | $LPIPS_{vgg} \downarrow$ |
> | --- | --- | --- | --- |
> | 100 | $31.39$ | $0.969$ | $0.041$ |
> | 2000 | $31.35$ | $0.968$ | $0.037$ |
> | 500 (Original) | $32.07$ | $0.971$ | $0.038$ |
>
> As shown, our model performance is not very sensitive to the point growing interval.
>
> We also show the results of using different numbers of top nearest neighbours (NN) to determine the sparse region in the table below.
>
> | Number of NNs | PSNR $\uparrow$ | SSIM $\uparrow$ | $LPIPS_{vgg} \downarrow$ |
> | --- | --- | --- | --- |
> | 5 | $31.54$ | $0.969$ | $0.041$ |
> | 20 | $31.58$ | $0.970$ | $0.040$ |
> | 10 (Original) | $32.07$ | $0.971$ | $0.038$ |
>
> As shown, our model performance is not very sensitive to the numbers of top nearest neighbours (NN) for determining the sparse region either.
>
> ### Q3: Is there a way to use the learned affinity to identify the sparse regions?
>
> A3: That is an interesting question. One idea is to use the entropy of the attention weights as a heuristic for identifying the regions for point growing. Specifically, if the attention weights have large entropy, it could mean that the model is less certain about which points to combine to produce the final colour. Therefore, more points may be helpful in that region.
>
> ### Q4: Could the proposed method refine the sparse reconstruction quality if we have a sparse reconstruction result for the scene by COLMAP? I am also curious about the potential of doing bundle adjustment with the proposed PAPR.
>
> A4: Yes, these are definitely possible and will be exciting avenues to explore in future works.

---

> > ### Comment · Reviewer_fb1M · 2023-08-19
> > **Response to the author rebuttal**
> >
> > I wish to express my appreciation for the authors' dedicated efforts in addressing my queries. I am satisfied with the responses provided and am impressed by that PAPR achieves high-quality point-based rendering without initialization.

---

> > > ### Author Response · Authors · 2023-08-19
> > > **Thank you for the response**
> > >
> > > Thank you for your response and recognition of our method. We're glad that our response addressed your concerns. If you have any more questions, please feel free to let us know.

---

### Official Review · Reviewer_QEnw · 2023-07-08

**Soundness:** 2 fair
**Presentation:** 3 good
**Contribution:** 2 fair
**Rating:** 7
**Confidence:** 4

**Summary:**

This paper tackles the problem of point-based rendering and proposes a new point-based differentiable rendering pipeline with proximity attention. With an attention-based point rendering, the proposed pipeline avoids the tuning of points' radius or splatting sizes that typical point rendering requires. Such a design enables efficient and high-quality rendering. The effectiveness is verified on NeRF synthetic and Tank and Temple datasets.

**Strengths:**

1. The proposed proximity attention is neat and effective.

1. The paper is well-written and easy to follow.

2. The experiments are solid.

**Weaknesses:**

### 1. Choices of query, key, value for attention

a. I think the paper misses an important ablation of various representations for query, key and value for the proximity attention. Currently, query is represented as ray direction, key is used as $(t, s, \mathbf{p})$, and value is represented as $(t, s, \mathbf{u})$. A natural question is what if 1) we remove $s$ in both key and value; 2) we have key and value are both represented as $(t, s, \mathbf{p}, \mathbf{u})$ (adding feature vector)?

b. What is related is Fig. 7. Can authors clarify: for "incrementally remove key components in the point embedding" (Fig. 7's caption), does it mean removing $t_{i,j}$ in both key and value representation?

### 2. Relationship to [24]

a.  I feel like the proximity attention is quite similar/related to the **Ray-centric Point Encoding** in [24]. Essentially, [24] also proposes an attention-based point embedding, where "query" is the ray direction same as this paper; "key" and "value" are concatenations from points's positions, distance between point and ray, angle between points and ray (also similar to the paper).

b. Therefore, to better position the paper, I want to understand the difference between this paper and [24]. This is one of the major motivations why I am curious about the ablations about the choices of query/key/values.

c. One difference I can see is that [24] uses an MLP as the renderer while PAPR uses a UNet. Can the author provide an ablation where an MLP is used for $f_{\theta_\mathcal{R}}$ (L181)? If PAPR does not work well with MLP, then it seems like the major improvement may come from the renderer's capabilities instead of the proposed attention-based point rendering.

d. Another difference is that [24] uses an image-based feature as the point's feature. Such a design does not allow for changing points's positions if the points's features are not updated later. For Fig. 4, authors state "we made modifications to their official code to enable point position optimization" (L214-215). Can the authors clarify whether the features will be updated or not? If the features are not updated, then this needs clarifications and it may be better to replace [24] results in Fig. 4 with initializing from a sparse point cloud of SfM or MVS. This will not shade the advantage of PAPR.

e. Similarly to d above, do the results of [24] in Tab. 1 come from initializing from a sphere? If so, I think it is better to replace it with initializing from a sparse point cloud and give a note in the table about this.

### 3. Speed

a. An advantage of traditional point-based rendering [15, 53] is the speed. Can authors also provide the inference speed of PAPR?

b. What is related is the training speed. Can authors also provide the training time of PAPR?

**Questions:**

See above.

**Limitations:**

The authors provide discussion of limitations in the paper.

---

> ### Author Rebuttal · Authors · 2023-08-10
>
> ### Q1: Additional ablation studies on key and value design for the proximity attention.
>
> A1: Sure. We conduct the ablation study on the key and value design using the NeRF Synthetic dataset.
>
> |  | PSNR | SSIM | LPIPS-vgg |
> | --- | --- | --- | --- |
> | No s in both K and V | 30.11 | 0.963 | 0.049 |
> | Use (s,t,p,u) for K and V | 30.91 | 0.968 | 0.041 |
> | Original K and V | 32.07 | 0.971 | 0.038 |
>
> As shown in the results above, our proposed combination achieves the best performance.
>
> ### Q2: How were the different point encoding components removed in the ablation study  (Fig. 7)?
>
> A2: We first remove p from the key, then we remove t in **both** key and value.
>
> ### Q3: Relationship to [24]
>
> A3: Indeed, our method is different from [24] in various ways, including point embedding design, renderer architecture and point feature.
>
> **Point Embedding Design:** Our point embedding is simpler than [24], and our experimental results validate the effectiveness of our method in learning the scene geometry and rendering high quality results.
>
> **Renderer Architecture:** We conduct the ablation study where we replace the UNet architecture for $f_{\theta_{\mathcal{R}}}$ with an MLP on the NeRF Synthetic dataset. The results are shown in the table below.
>
> | Renderer Architecture | PSNR | SSIM | LPIPS-vgg |
> | --- | --- | --- | --- |
> | MLP | 30.54 | 0.962 | 0.054 |
> | U-Net (Original) | 32.07 | 0.971 | 0.038 |
>
> As shown above, our model with the modified renderer architecture still significantly outperforms [24] in terms of all metrics. This validates the efficacy of our method does not solely come from the renderer’s architecture. Nevertheless, we do find that the UNet architecture provide a bit of improvement to the rendering quality in our case.
>
> **Point Feature:** As the reviewer pointed out, another distinction is [24] used image-based features, whereas our point features are view-independent and do not depend on images. Yes, for [24], we updated the features jointly with the point coordinates, including for the results in Fig. 4. Despite this modification, [24] failed to capture the scene geometry correctly.
>
> **Whether [24] was initialized with the original initialization method:** Actually, for all the results in Tab. 1, we ran the baseline methods with their original point initialization methods. In contrast, our method was initialized from a random point cloud. So we gave all baselines an advantage over our method, which makes the evaluation setting more challenging. In particular, the point positions of [24] are initialized using a sparse point cloud obtained from COLMAP.
>
> While it was not feasible to study and evaluate the effect of every facet of our design due to resource limitations, we have included our code with this submission to facilitate a thorough exploration of the finer details by reviewers.
>
> ### Q4: Can authors also provide the inference speed of PAPR?
>
> A4: Certainly. To render an 800x800 image, our model takes 2.8s, while NeRF would take 11.8s. Our code's inference speed is not yet optimized, with the standard PyTorch topk function constituting about 96% of total inference runtime. Utilizing custom CUDA kernels could potentially accelerate inference speed further.
>
> ### Q5: Can authors also provide the training time of PAPR?
>
> A5: Of course. It takes about 20 hours to train a model for a scene from either NeRF Synthetic or Tanks & Temples datasets. In comparison, NeRF would require 20 hours to train on a scene from the NeRF synthetic dataset, and 30 hours on a scene from Tanks & Temples subset.

---

> > ### Comment · Reviewer_QEnw · 2023-08-18
> > **Rebuttal Reply**
> >
> > I appreciate authors's time and effort in addressing my questions.
> >
> > My major concern about the relationship with [24] gets resolved and I am happy with the additional information provided. I think this is solid work and I stay positive on the acceptance. Therefore, I raise my score.

---

> > > ### Author Response · Authors · 2023-08-19
> > > **Thank you for the response**
> > >
> > > Thank you for responding and updating your rating. We're glad to hear that your major concerns have been resolved and that the additional information we provided was helpful. If you have any further questions, please feel free to let us know.

---

### Author Rebuttal · Authors · 2023-08-10

# General Responses

We thank all reviewers for dedicating their time, providing insightful feedback, and unanimously acknowledging the efficacy of the proposed method and the good results. In particular, the reviewers remarked, "the proposed proximity attention is neat and effective”(R QEnw), "the overall pipeline is neat and technically sound." (R fb1M), "the representation is simple, reasonable and well-designed." (R dttq), “Solid pipeline, great results, good demonstrations and presentation” (R TJqY), “The visualization is very interesting and insightful, the proposed components are thoroughly evaluated and the improvement are significant” (R U5yB).

We found the comments and questions very helpful, which add value to our work. Here, we provide a one-sentence summary of our response to selected questions raised by the reviewers — please refer to our individual responses to each review for the details.

### Q1: Please visualize the attention weights. Does the learned renderer actually utilizes the information of $s_{i, j}$ and $t$ as we expect? (R ****TJqY, U5yB)****

A1: Yes, the attention weight visualization in the rebuttal PDF does indeed show higher weights for closer points.

### Q2: Is feature vector learned or deterministically determined? (R ****dttq)****

A2: The feature vector is learnt as mentioned in L207 - 208.

### Q3: How can this approach be extended to non-rigidly deforming scene? (R ****dttq)****

A3: Yes. In fact, we’ve showcased a non-rigid stretching deformation of the mic in the third column of Figure 8a.

### Q4: How were the different point encoding components removed in the ablation study  (Fig. 7)? (R ****QEnw)****

A4: We first remove p from the key, then we remove t in **both** key and value.

### Q5: Only one attention layer? (R ****TJqY)****

A5: Yes.

### Q6: Computational complexity of attention per ray. (R ****U5yB)****

A6: The complexity is O(K), where K is the k parameter in the top-k operation as described on L164-166.

### Q7: Do you need special initialization for the MLPs? (R ****U5yB)****

A7: No. The weights to the MLPs are randomly initialized.

---

### Decision · Program_Chairs · 2023-09-21

**Decision:**

Accept (spotlight)

**Comment:**

A lot of support for this work. ``U5yB`` was still slightly negative, but their concerns are not perceived as blocking by the AC. AC tends to understand what the paper proposes and is supportive, too.

A small observation not related to or affecting this decision: Authors were quite pushy, asking reviewers and AC to respond. Please could you consider abstaining from this next time? It just creates a lot of noise in an already stressful process, thanks!